# Design of synthetic human gut microbiome assembly and butyrate production

Ryan L. Clark [1], Bryce M. Connors[1,2], David M. Stevenson[3], Susan E. Hromada [1,3], Joshua J. Hamilton[1], Daniel Amador-Noguez [3] & Ophelia S. Venturelli [1,2,3✉]

The capability to design microbiomes with predictable functions would enable new technologies for applications in health, agriculture, and bioprocessing. Towards this goal, we develop a model-guided approach to design synthetic human gut microbiomes for production of the health-relevant metabolite butyrate. Our data-driven model quantifies microbial interactions impacting growth and butyrate production separately, providing key insights into ecological mechanisms driving butyrate production. We use our model to explore a vast community design space using a design-test-learn cycle to identify high butyrate-producing communities. Our model can accurately predict community assembly and butyrate production across a wide range of species richness. Guided by the model, we identify constraints on butyrate production by high species richness and key molecular factors driving butyrate production, including hydrogen sulfide, environmental pH, and resource competition. In sum, our model-guided approach provides a flexible and generalizable framework for understanding and accurately predicting community assembly and metabolic functions.

[1] Department of Biochemistry, University of Wisconsin-Madison, Madison, WI, USA. [2] Department of Chemical & Biological Engineering, University of Wisconsin-Madison, Madison, WI, USA. [3] Department of Bacteriology, University of Wisconsin-Madison, Madison, WI, USA. ✉email: venturelli@wisc.edu

Microbial communities carry out pivotal chemical transformations in nearly every environment on Earth[1]. Many of these processes critically impact human health, agriculture and environmental sustainability, including oceanic $CO_2$-fixation[2], production of growth-promoting molecules in the plant rhizosphere[3], and degradation of indigestible dietary substrates in humans[4]. The functions performed by microbiomes are shaped by complex and dynamic interactions between constituent community members and their environment. Developing the capabilities to engineer microbiome properties holds promise to address grand challenges facing human society in agriculture, degradation of the environment, and human health[5].

Undefined communities isolated from natural environments have been exploited to treat *Clostridioides difficile* infection in the human gut[6], produce valuable chemicals in anaerobic digestors[7], or extract minerals via bioleaching[8]. Optimization of such desired functions performed by undefined communities is challenging due to unknown components and their interactions and the difficulty in quantifying the contributions of organisms to the function of interest. Further, undefined communities used in fecal microbiota transplantation (FMT) to treat human diseases have resulted in unpredictable patient outcomes attributed to complex and unknown ecological interactions between donor and recipient microbiomes[9]. In addition, FMT can lead to potential adverse health consequences including the transfer of antibiotic-resistant pathogens[10–12]. For these reasons, there is substantial interest in exploiting the properties of defined microbial communities for enhanced controllability, reproducibility, and environmental safety.

The bottom-up construction of synthetic microbiomes enables the investigation of reduced complexity assemblages with control of initial community composition (organism type and initial abundance of constituent community members). Defined microbial communities have been extensively studied both in vitro[13] and in model host systems[14,15] to decipher potential causal interactions influencing system behaviors. Using in vitro culturing, synthetic communities of bacteria isolated from the human gut[16] or soil[17] have been used to demonstrate that pairwise interactions are major determinants of multi-species community assembly. Beyond community assembly, synthetic communities have been used to interrogate how community context impacts the production or consumption of molecules relative to individual species[18–20]. In real-world environments, defined consortia have been used for environmental cleanup[21], protection of plant hosts from pathogen infection[22,23], and treatment of recurrent *C. difficile* infection[24,25]. However, the widespread deployment of synthetic microbiomes has been hindered by our limited capabilities for accurate prediction of community assembly and function.

Identifying synthetic communities that perform functions with desired specifications is highly challenging due to a large design space that grows exponentially with the number of organisms. The web of interactions between constituent community members and abiotic factors (e.g. nutrients and environmental pH) can potentially generate many local optima representing a rugged community composition–function landscape[26]. In addition, the community composition–function landscape can be shaped by variation in the activity of metabolic pathways (i.e. metabolic flexibility) driving the function of interest due to the availability of ecological niches[27]. Therefore, developing the capabilities to systematically explore this complex landscape could enable the discovery of synthetic microbiomes with optimized functions that can be harnessed for diverse applications.

In metabolic and protein engineering, computational models informed by high-throughput experimental measurements have been used to design optimized protein sequences with high protein stability[28] or identify promoter combinations driving enzymes in a metabolic pathway to maximize valuable molecule production[29,30]. In the microbiome field, empirical and top-down approaches are frequently used to design microbial communities as opposed to data-driven approaches[14]. In one example, a model-guided procedure was used to identify gut microbial communities that elicit a target immune response in mouse models[31]. However, variation in host colonization masked the contributions of specific microbes and the high biological variability between replicates of the same community limited the observability of significant increases in the immune response. Computational modeling frameworks that can predict how interactions between species shape community assembly and functional outputs are needed to advance the rational design of synthetic microbiomes.

Butyrate production is a major function of the gut microbiome associated with protection from a wide range of human diseases, including arthritis[32], diet-induced obesity[33–35], colitis[36,37], opportunistic pathogen infection[38], diabetes[39], and colorectal cancer[40]. Predictable modulation of butyrate production of gut microbiota with defined bacterial therapeutics is challenging as evidenced by the failure to modulate fecal butyrate concentration in individuals with metabolic syndrome via supplementation with a butyrate producing bacterial strain[41]. The effects of introducing invader therapeutic species depends on the availability of ecological niches and the complex interactions between resident and invader species. A detailed and quantitative understanding of these interactions could guide the design of microbiome interventions for predictable shifts in butyrate concentration.

Due to significant interest in the development of defined bacterial therapeutics for human health applications[42] and the beneficial role of butyrate produced by gut microbiota on myriad health outcomes[32–41], we sought to develop the capability to predict butyrate production of diverse synthetic human gut communities. To this end, we leveraged interpretable data-driven models to design synthetic communities with desired butyrate concentrations and to decipher significant microbial interactions impacting butyrate production. By quantifying interactions that impact growth and butyrate production separately, we demonstrated that in some contexts, accurate prediction of butyrate producer abundance can predict butyrate production, while in other community contexts, interactions modifying metabolic activities must be captured. We used our models to design communities of up to 25 diverse and prevalent gut bacterial species with a broad range of butyrate production capabilities and uncovered key insights into metabolic interactions impacting butyrate production by analyzing our model as well as the metabolic profiles and environmental modification of the designed communities. Finally, we used our model to investigate how biodiversity–ecosystem function relationships constrain the design of butyrate production in the system. Our generalizable framework provides a foundation for the data-driven design of complex microbial communities for biological objectives and could be extended to other microbiome engineering applications including the production or degradation of health-relevant molecules, production of compounds to enhance plant growth, or sustainable biomanufacturing of valuable molecules.

## Results
**Model-guided procedure guides the exploration of butyrate production landscapes.** We aimed to explore the butyrate production landscape as a function of community composition to decipher microbial interactions shaping butyrate production. Exploring the butyrate production functional landscape is a major challenge because the number of sub-communities increases

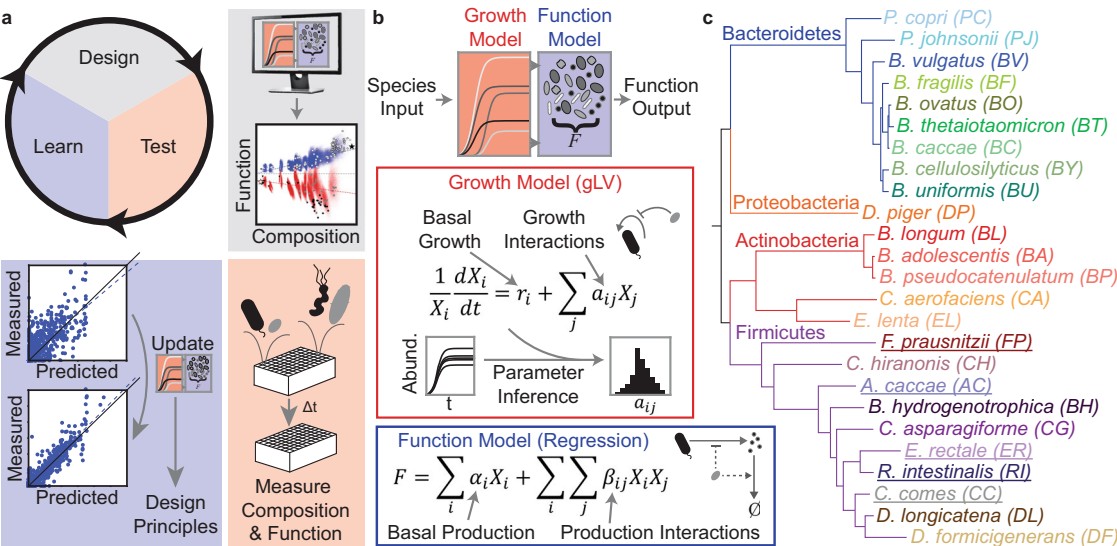

**Fig. 1 Iterative modeling framework to predict microbial community assembly and metabolic function. a** Two-stage modeling framework for predicting community assembly and function. The generalized Lotka–Volterra model (gLV) represents community dynamics. A Bayesian Inference approach was used to determine parameter uncertainties due to biological and technical variability. A linear regression model with interactions maps assembled community composition to metabolite concentration. Combining these two models enables prediction of a probability distribution of metabolite concentration from initial species abundances. **b** Design–Test–Learn cycle for model development. First, we use our model to explore the design space of possible experiments (i.e. different initial conditions of species presence/absence) and design communities that span a desired range of metabolite concentrations. Next, we use high-throughput experiments to measure species abundance and metabolite concentration. Finally, we evaluate the model's predictive capability and infer an updated set of parameters based on the new experimental measurements. **c** Phylogenetic tree of the synthetic human gut microbiome composed of 25 highly prevalent and diverse species. Branch color indicates phylum and underlined species denote butyrate producers.

exponentially with the number of species[43]. To investigate the landscape, we developed a modeling framework to guide the iterative design of informative experiments (Fig. 1a, b). Microbial interactions can impact growth or metabolite production by influencing the availability of ecological niches or facilitating metabolite degradation. To capture these two types of interactions, we implemented a two-stage modeling framework to determine the contributions of microbial interactions to species growth and community assembly or metabolite production. In the first stage, a dynamic ecological model, referred to as the generalized Lotka–Volterra model (gLV), predicts community assembly. The second stage predicts metabolite production as a function of the resulting community composition (Fig. 1b). The gLV model is a set of coupled ordinary differential equations that capture the temporal change in species abundances due to monospecies growth parameters and inter-species growth interactions (see the "Methods" section)[16]. To estimate parameters for the gLV model, we use Bayesian parameter inference techniques to determine the uncertainty in our parameters based on biological and technical variability in the experimental data[44].

Our metabolite production model consists of a linear regression model with interaction terms mapping community composition (i.e. abundance of each species) at a specific time point to the concentration of an output metabolite at that time. This model was based on a phenomenological model of metabolite production used in bioprocess engineering expanded to microbial communities (see the "Methods" section). In the regression model, the first-order terms capture the monospecies production per unit biomass and the interaction terms represent the impact of inter-species interactions on metabolite production per unit biomass (i.e. deviations from constant metabolite production per unit biomass[19]). To estimate parameters for the regression model, we use Lasso regression to identify the most impactful interactions. Altogether, the composite gLV and regression model predicts the probability distribution of the

metabolite concentration given an initial condition of species abundances (Fig. 1b, see the "Methods" section).

In metabolic and protein engineering, a design–test–learn cycle (DTL) has been used to design biomolecules[45] or metabolic pathways[46] with properties that satisfy desired performance specifications. We hypothesized that this engineering-inspired approach could be used to explore community design spaces and understand the composition–function mapping for butyrate production. Each cycle consisted of: (1) a design phase wherein we used our model informed by experimental observations to simulate a vast number of potential community compositions to identify sub-communities that satisfied biological objectives (i.e. desired butyrate concentrations), (2) a test phase wherein the selected sub-communities were assembled and species abundance and butyrate concentration were measured, and (3) a learn phase wherein patterns in our experimental data were used to estimate model parameters and to extract information about the key microbial interactions influencing community assembly and butyrate production.

**Two-stage model enables efficient exploration of low richness community design space.** To develop a system of microbes representing major metabolic functions in the gut, we selected 25 prevalent bacterial species from all major phyla in the human gut microbiome[47] (Fig. 1c, Supplementary Data 1). This community contained five butyrate-producing Firmicutes which have been shown to play important roles in human health and protection from diseases (Fig. 1c, Supplementary Data 1). Due to the lack of a defined medium that universally supports the growth of gut microbes, most in vitro studies use undefined media, making it difficult to interrogate the effects of unknown components on community behaviors[48]. To maximize our knowledge of the substrates available to the communities, we developed a chemically defined medium to grow the synthetic communities (see the "Methods" section).

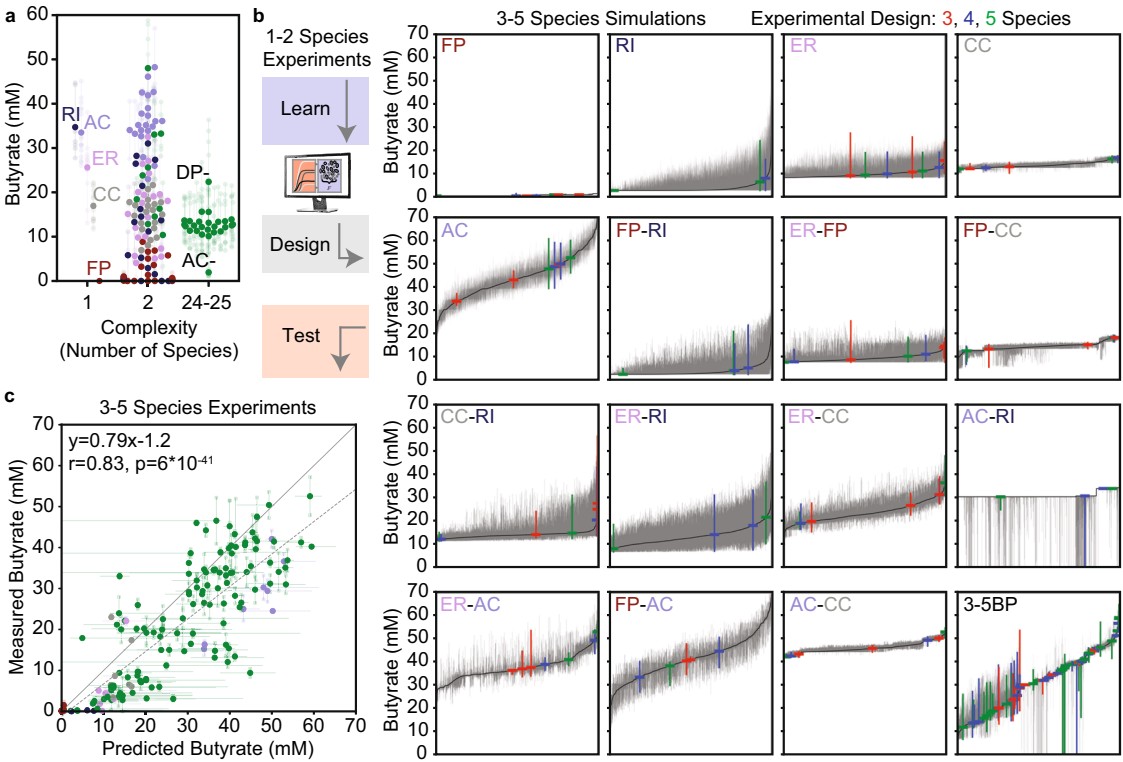

**Fig. 2 Exploring the predicted butyrate production of 3–5 member communities with a model trained on 1–2 species communities. a** Categorical scatter plot of butyrate production in 1–2 species and 24–25 species communities. Solid datapoints indicate the mean of the biological replicates which are represented by transparent datapoints connected to the mean with transparent lines. The colors indicate which butyrate producer was present in the community with green indicating the presence of multiple butyrate producers. DP− and AC− indicate the 24-species communities lacking *Desulfovibrio piger* (DP) and *Anaerostipes caccae* (AC), respectively. **b** Predicted medians (black line) and 60 percent confidence intervals (gray bars) of butyrate concentration for all 3–5 member communities containing at least one butyrate producer (46,591 community predictions). Colored lines indicate median and 60 percent confidence interval of butyrate production of communities chosen for the experimental design with the color indicating the number of species in the community (156 communities). Subplots separate groups of communities based on the identities of the combination of butyrate producers specified. **c** Scatter plot of measured butyrate versus predicted butyrate for 3–5 species communities. Colors indicate which butyrate producer was present in the community as in **a**. Biological replicates ($n = 1$–5, depending on the community, exact values in source data) are indicated by transparent squares connected to the corresponding mean, which is represented by the large data point. Prediction error bars (*x*-axis) indicate the 60% confidence interval of the predicted butyrate distribution as in **b**, with the center being the median prediction. Dashed line indicates the linear regression between the mean measured butyrate and the median predicted butyrate. Indicated statistics are for Pearson correlation (two-sided). Source data are available in the Source Data file.

Based on previous studies using pairwise communities to predict higher richness community behaviors[16,18,49], we hypothesized that training our model on single and pairwise community measurements would provide an informative starting point for mapping composition–function relationships determining butyrate production. To do so, we first measured time-resolved growth of single species and observed a wide variety of growth dynamics within each phylum, including disparate growth rates and carrying capacities (Supplementary Fig. 1). We assembled each pairwise community containing at least one butyrate producer (the focal species of our system[50]) and measured species abundance and the concentrations of organic acid fermentation products (including butyrate, lactate, succinate, and acetate) after 48 h. The pairwise consortia displayed a broad range of butyrate concentrations of 0–50 mM (Fig. 2a).

Single-species deletion communities have been used to investigate the contributions of individual species to a community function[13,16]. Therefore, we characterized the full 25-species community and each single-species deletion sub-community (i.e. 24-member consortia). In stark contrast to the pairwise communities, the 24- and 25-species communities exhibited

similar low butyrate production (~2–22 mM Butyrate). The absence of only two species *Desulfovibrio piger* (DP) (~22 mM Butyrate) and *Anaerostipes caccae* (AC) (~2 mM Butyrate) resulted in a significant increase or decrease in butyrate concentration compared to the remaining 24-member and 25-member communities (Fig. 2a, Supplementary Fig. 2a). In addition, the concentrations of all measured organic acids spanned a much smaller range in the 24 and 25-member communities than the single and pairwise consortia (Supplementary Fig. 2b). These results suggest that high richness communities may trend towards a similar low butyrate-producing state that is difficult to change by the deletion of most single species and motivates a model-guided design strategy for exploring how community richness shapes butyrate production.

To determine whether individual and pairwise communities could predict community composition and butyrate production of low richness communities (i.e. 3–5 species), we estimated the parameters of our model based on experimental measurements. Our initial model was informed only by pairwise communities that contained at least one butyrate producer (Supplementary Data 2, M1) and was thus naïve to all interactions between non-butyrate

producers. We assumed that the unobserved growth interactions could be predicted based on trends in measured interactions across phylogenetic relatedness (see the "Methods" section)[16]. However, the resulting model was unable to predict butyrate production in the 24-and 25-member communities (Supplementary Fig. 3), which we attributed to missing information about non-butyrate producer interactions in our training data. Thus, we used our model to explore a low richness design space of 3–5 species communities based on the assumption that pairwise interactions would be more observable in low than high richness (i.e. >10 species) communities to identify an improved parameter estimate for non-butyrate producer interactions.

We used our initial M1 model to predict the probability distributions of butyrate production for all 3–5 species communities containing at least one butyrate producer (46,591 communities). The predicted butyrate production varied substantially based on the combination of butyrate producers present in each community (Fig. 2b). In addition, we observed variations in the shapes of the probability distributions based on how the uncertainty in growth prediction propagated through the regression model. For instance, the butyrate concentration in the AC, *Roseburia intestinalis* (RI) pairwise community was lower than the AC monoculture, even though RI was low abundance, resulting in a high magnitude negative parameter in the regression model for a production interaction between AC and RI (Supplementary Data 3). Due to the uncertainty in the growth parameters, the model predicted that RI would grow substantially in a subset of the 3–5 member simulations containing both AC and RI. The variability in predicted RI growth combined with the high magnitude negative interaction parameter between AC and RI resulted in distributions where the median butyrate concentration was high (i.e. for simulations where RI did not grow substantially), and the 60 percent confidence interval extended to 0 mM butyrate (i.e. when RI grew substantially) (Fig. 2b). In sum, these results demonstrate that the shape of the predicted probability distributions can provide information about the uncertainty in species growth based on experimental observations.

Based on the simulations, we selected 156 communities that spanned a broad range of predicted butyrate concentrations across the butyrate producer groups to evaluate experimentally (Fig. 2b). The model prediction exhibited good agreement with the rank order of butyrate production (Spearman rho = 0.84, $p = 9*10^{-43}$) (Fig. 2c) and species abundance (Spearman rho = 0.76, $p = 3*10^{-122}$) (Supplementary Fig. 4a–d), demonstrating that our initial model could predict a wide range of butyrate production in low richness communities.

**Composition–function landscape predicts contributions of growth and production interactions.** Encouraged by our model's predictive ability, we sought to explore composition–function relationships in higher richness communities (i.e. >10 species) using a model with updated parameters based on measurements of the 3–5 member communities (Supplementary Data 2, M2). Since the human gut microbiome exhibits functional redundancy in butyrate pathways[51], we first used model M2 to simulate the assembly of all communities containing all five butyrate producers (5-butyrate producer or 5BP, 1,048,576 total) to map the composition–function landscape for butyrate production (Fig. 3a). In addition, we simulated the assembly of all communities containing the four butyrate producers excluding AC (4-butyrate producer or 4BP, 1,048,576 total) to understand how the composition–function landscape changes in the absence of the most productive butyrate producer (Fig. 3b). The majority of 5BP communities were predicted to have higher butyrate

concentration than any of the 4BP communities (Fig. 3a, b), consistent with the substantial decrease in butyrate in the AC deletion community observed previously (Fig. 2a).

The relationship between butyrate producer abundance and butyrate can provide insight into the contributions of growth and production interactions in the presence and absence of AC (Fig. 3a, b). If butyrate producer abundance correlates with butyrate, then growth interactions drive butyrate production, whereas the contributions of production interactions would reduce the strength of this correlation. The 5BP communities were predicted to have a large contribution of production interactions as evidenced by a weak correlation between butyrate concentration and butyrate producer abundance (Spearman rho = 0.17, $p < 0.0001$) (Fig. 3a). By contrast, the predicted butyrate producer abundance and butyrate concentration exhibited a stronger correlation in 4BP communities (Spearman rho = 0.57, $p < 0.0001$), suggesting that growth interactions dominated butyrate production (Fig. 3b). The 5BP communities were partitioned into low and high butyrate production clusters defined by the presence or absence of DP due to strong negative production interactions between DP and the butyrate producers (Fig. 3a), consistent with the increase in butyrate concentration in the DP deletion community compared to the full community (Fig. 2a).

To validate our model's prediction of the composition–function landscapes of the 4BP and 5BP communities, we designed a subset of the communities to test experimentally by focusing on communities that deviated from the average butyrate production. Thus, for the 5BP landscape, we designed 28 low- and 54 high-butyrate communities each with 11–17 species. For comparison with the designed set, we randomly selected 82 communities with the same range of species richness (Fig. 3a). The 82 model-designed communities exhibited a higher variance in mean butyrate production than the 82 randomly selected communities (designed communities s.d. = 11 mM, random communities s.d. = 8 mM, Levene test, $p = 0.043$) (Fig. 3a, c), demonstrating a major advantage of the model-guided DTL approach for designing communities that span a wide range of functional output levels. In addition, these communities exhibited no correlation between butyrate producer abundance and butyrate concentration, consistent with the predicted weak correlation (Fig. 3a, c) and demonstrating the major role of production interactions in 5BP communities. Notably, many of the high richness communities produced substantially higher butyrate than the 5-butyrate producer community alone, highlighting an additional advantage of our model to identify highly functional communities in the vast composition–function landscape. Further, these results demonstrate that organisms that do not directly contribute to butyrate production can still have a major contribution by modifying butyrate producer growth or production. In our system, AC has the unique capability to transform lactate to butyrate[52] in addition to production of butyrate from sugars (Supplementary Fig. 5a), suggesting that the ability to use an alternative metabolic pathway depending on nutrient availability may be a key determinant of the contribution of production interactions.

The set of 84 4BP communities containing 11–19 species exhibited a very strong correlation between butyrate producer abundance and butyrate concentration (Spearman rho = 0.78, $p = 8*10^{-19}$), consistent with the strong correlation predicted by our model (Fig. 3d). By contrast with the 5BP communities, the 4-butyrate producer community exhibited higher butyrate than all of the designed high richness communities (Fig. 3d), demonstrating that the non-butyrate producers inhibited butyrate production in the absence of AC, primarily through growth

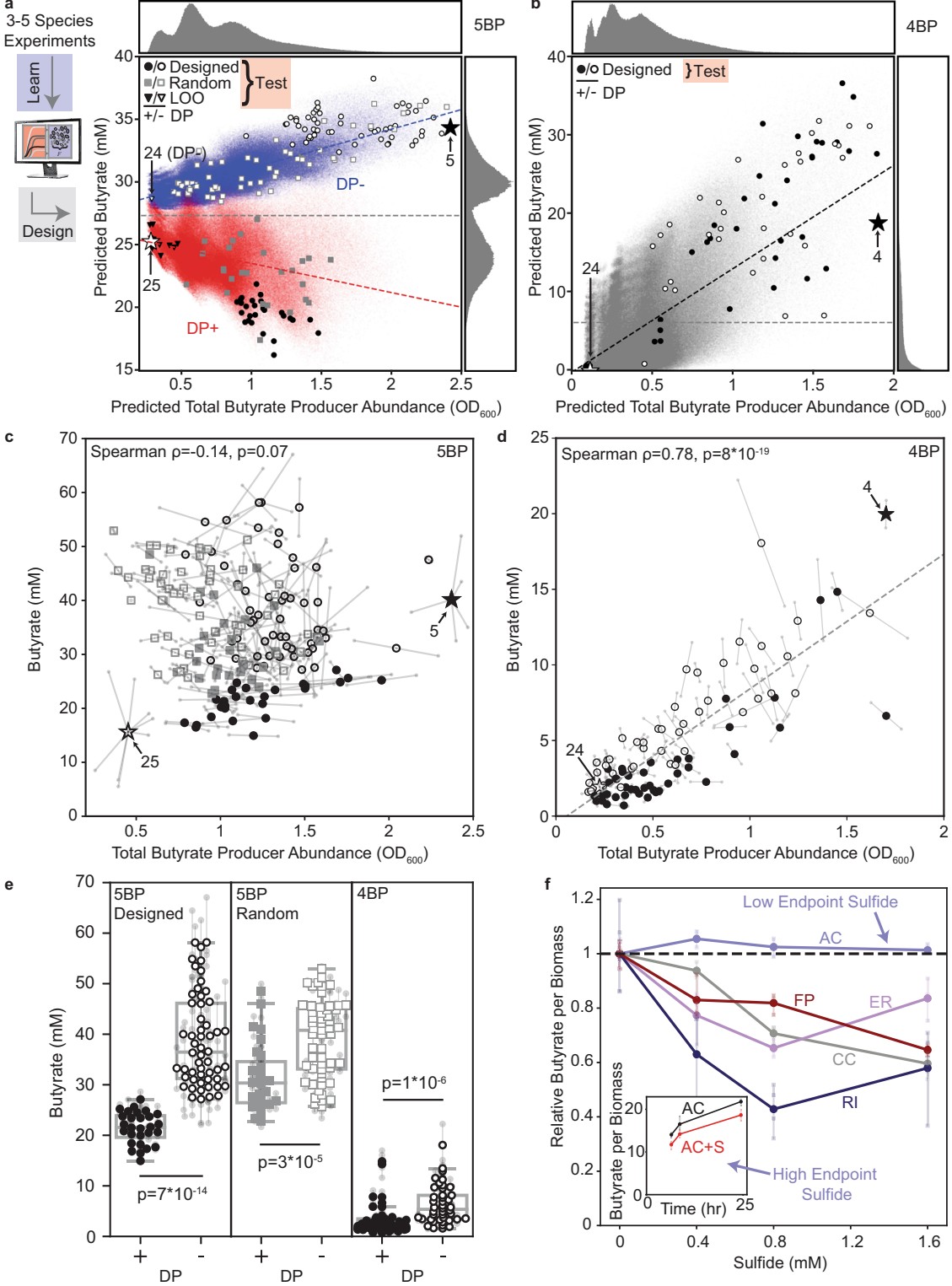

interactions. In sum, our model was able to decipher the contributions of growth and production interactions in high-richness communities (i.e. >10 species) based on measurements of lower-richness communities (i.e. 1–5 species). Further, these results demonstrate that the presence of a single species can transform the composition–function landscape by shifting the type of interactions driving a metabolic function between growth and production.

**Hydrogen sulfide production by DP inhibits butyrate production.** Consistent with our model predictions, communities containing DP had lower butyrate concentration than those without DP in both the designed and randomly selected 5BP communities and the 4BP communities (Fig. 3e). Our model parameters suggested that this was due to a negative production interaction between DP and the butyrate producer AC

**Fig. 3 Community composition–function landscapes reveal key role of production interactions on *A. caccae* and negative impact of *D. piger* on butyrate production. a** Scatter plot of predicted total butyrate producer abundance versus predicted butyrate concentration for all possible communities in which all five butyrate producers are present (1,048,576 communities). Histograms indicate the butyrate concentration distribution across the given axis. Communities are colored according to the presence (red) or absence (blue) of *D. piger* (DP). Blue and red dashed lines indicate the linear regression of communities with (red, $y = -2.3x + 25.8$, $r = -0.34$) or without (blue, $y = 3.1x + 28.0$, $r = 0.76$) DP. The white star indicates the full 25-member community and black star indicates the community of five butyrate producers alone. Large data points indicate communities chosen for experimental validation. Black triangles indicate leave-one-out communities, black circles indicate designed communities, and gray squares indicate random communities, with open/closed symbols indicate the absence/presence of DP. **b** Scatter plot of predicted total butyrate producer abundance versus predicted butyrate concentration for all possible communities in which all four butyrate producers excluding AC are present (1,048,576 communities). Histograms indicate the butyrate concentration distribution. Gray dashed line indicates the mean predicted butyrate concentration across all communities. Black dashed line indicates the linear regression of all communities ($y = 13.2x - 0.1$, $r = 0.64$). The white star indicates the full 24-member community and the black star indicates the four butyrate producers alone. Large data points indicate communities chosen for experimental validation. **c, d** Scatter plots of experimental measurements of total butyrate producer abundance versus butyrate concentration for communities with (**c**) and without (**d**) AC. Data point shapes correspond to the legends in (**a**) and (**b**) and represent the mean of biological replicates, which are shown as small datapoints connected to the corresponding mean with lines ($n = 2$ except for 5 BP, 24 and 25 species communities where $n = 5$–8). Dashed line in (**d**) indicates the linear regression ($y = 8.9x - 0.5$). **e** Comparison of butyrate concentration in communities from **a** and **b** with and without DP for both the designed and random experimental sets for the 5BP communities and designed set for the 4BP communities. Data point shapes correspond to the legends in **a** and **b** and represent the mean of biological replicates, which are shown as small datapoints connected to the corresponding mean with lines. Box and whisker plots represent the median (center line), quartiles (box), and range (whiskers) of the mean butyrate concentration for each community, excluding outliers (points outside 1.5 times the interquartile range). Indicated *p*-values are from a Mann–Whitney *U* test (5BP Designed: $n = 28$ for DP+ and $n = 54$ for DP−; 5BP Random: $n = 27$ for DP+ and $n = 55$ for DP−; 4BP: $n = 42$ for DP+ and $n = 42$ for DP−). **f** Butyrate concentration per unit biomass as a function of sulfide concentration after 24 h of growth. Butyrate concentration per biomass was normalized to the no sulfide condition. Circles indicate the mean of biological replicates, with individual replicates shown as transparent squares ($n = 4$). Inset: Butyrate concentration per biomass (mM $OD_{600}^{-1}$) for AC with and without the addition of 1.6 mM sulfide across time ($n = 3$). Endpoint sulfide concentrations were higher in the data shown in the inset than in the main figure (Supplementary Fig. 6). Source data are available in the Source Data file.

(Supplementary Data 3, Model M2). Thus, we sought to investigate the molecular basis of this interaction.

Since AC was a highly productive butyrate producer (Fig. 2a), we first considered resource competition between AC and DP. While AC and DP have previously been shown to compete for lactate in vitro[53], excess lactate was present in communities containing both DP and AC, suggesting that competition over limited lactate was not a major determinant of the negative production interaction (Supplementary Fig. 5b).

Since some *Desulfovibrio* species have the capability to use butyrate as an energy source[54], we tested whether decreased butyrate in the presence of DP could be due to butyrate consumption. To investigate this hypothesis, we grew DP in media supplemented with different concentrations of sodium butyrate ranging between 0 and 100 mM and measured the butyrate concentration after 48 h of incubation. The presence of DP did not alter the concentration of butyrate in any condition relative to media controls, indicating that consumption or degradation of butyrate by DP was not a major factor contributing to the negative production interaction (Supplementary Fig. 5c).

One unique metabolic characteristic of DP in our system is the capability to reduce sulfate to hydrogen sulfide ($H_2S$). We next tested if $H_2S$ contributed to the negative impact of DP on butyrate production. To test this hypothesis, each butyrate producer was grown in media supplemented with sulfide in a range of concentrations similar to those observed in DP monoculture. Notably, the butyrate concentration per unit biomass decreased with increasing sulfide concentration for CC, FP, ER, and RI (Fig. 3f). For AC, butyrate concentration per unit biomass was not substantially different in the presence and absence of sulfide, but we observed substantially lower sulfide concentration at the end of the experiment, which we attributed to accelerated loss of $H_2S$ to the gas phase because of bubbling associated with AC's fast growth (Supplementary Fig. 6a–e). Thus, we grew AC with addition of 1.6 mM sulfide with sealed tubes to minimize loss of $H_2S$, and performed time-series measurements of butyrate concentration, sulfide concentration, and biomass. Our results showed lower butyrate production per unit biomass relative to the

sulfide-lacking control (Fig. 3f), suggesting that this trend was due to higher concentrations of sulfide maintained until later time points (Supplementary Fig. 6f). Our results suggest that $H_2S$ production by DP contributes to reduced butyrate concentration in communities. In addition, these data suggest that $H_2S$ produced by the host and constituent members of the gut microbiota could shape butyrate production in the human gut microbiome.

**Model informed by high richness communities can accurately predict butyrate production.** While model M2 elucidated global trends in the composition–function landscapes and identified the key negative production interaction by DP, the model required additional information to be quantitatively predictive of high richness communities (Supplementary Figs. 4e–h and 7). Therefore, we sought to continue the DTL paradigm by updating the parameters of our model with new measurements of high richness communities. We updated our model parameters by training on a subset of high richness communities: the random set of 5BP communities (82 communities) and a randomly sampled half of the 4BP communities (42 communities) (Supplementary Data 2, M3). Notably, the updated model M3 accurately predicted the butyrate concentration (Spearman rho = 0.86, $p = 9*10^{-45}$) and species abundances (Spearman rho = 0.82, $p < 0.0001$) of the remaining high richness communities, demonstrating that our model could explain the quantitative trends in the high richness communities when provided with sufficiently informative training data (Fig. 4a, Supplementary Fig. 4i–l).

We next sought to identify key microbial interactions impacting species growth using the inferred model parameters using our predictive model M3 (Fig. 4a). Negative interactions ($<-0.05$ h$^{-1}$ ($OD_{600}$ Species j)$^{-1}$) dominated the network, representing 49.8% of the interspecies interaction parameters (Supplementary Fig. 4j). By contrast, 1.7% of interactions were positive ($>0.05$ h$^{-1}$ ($OD_{600}$ Species j)$^{-1}$) (Supplementary Fig. 4j), consistent with previous observations of the prevalence of negative interactions in microbial communities[16,55]. To understand how inter-species growth interactions vary across chemical composition contexts, we compared the inferred inter-species

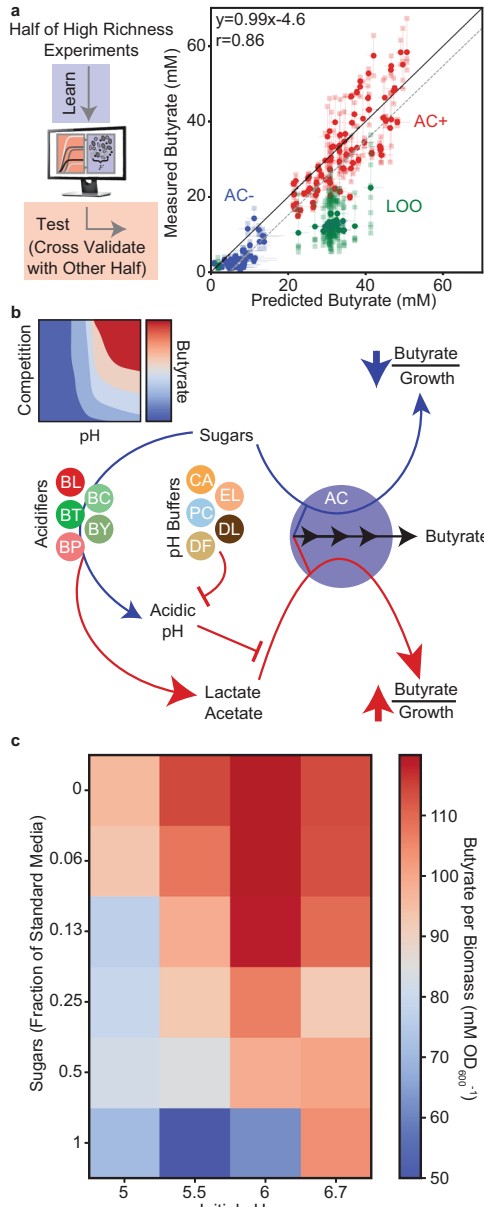

**Fig. 4 Model accurately predicts butyrate production and provides insight into the combinatorial effects of pH and resource competition on butyrate production of *A. caccae*. a** Scatter plot of predicted versus measured butyrate concentration for high richness communities. Model parameters were updated based on half of the high richness community data and used to predict the remaining half. Transparent data points indicate biological replicates and are connected to the corresponding mean by transparent lines ($n = 2$–8, depending on the community). Solid gray line indicates $x = y$. Dashed line indicates linear regression of prediction versus mean measurement ($y = 0.99x - 4.6$, Pearson correlation two-sided $r = 0.86$, $p = 1*10^{-44}$). **b** Schematic representing proposed relationship between pH and sugar competition in modulating butyrate production of AC in high richness communities. Red edges denote processes that negatively impact butyrate production and blue edges represent processes that enhance butyrate production. The abundance of "Acidifiers" were positively correlated with lactate concentration and negatively correlated with pH in high richness communities. The abundance of "pH Buffers" were positively correlated with pH in high richness communities. Note that species contributions to these processes are expected to be context-dependent. Heatmap represents proposed qualitative butyrate landscape as a function of the strength of resource competition for sugars and environmental pH. **c** Heatmap of AC monoculture butyrate yield per biomass with varying initial pH and sugar concentrations. Values indicated are the mean of three biological replicates for each condition. Source data are available in the Source Data file.

set includes high richness communities[43]. Based on this result, we hypothesized that an effective experimental design for predicting high richness communities would involve measurements of randomly selected high richness communities in lieu of the pairwise communities. To test this hypothesis, we constructed a new model based on the measurements of single species and the random set of 5BP communities (Supplementary Data 2, M4, 82 communities) and compared the predictive capabilities of model M4 to model M1, which was trained on single species and pairwise communities (110 pairwise communities). While these models were trained on a similar number of communities, M4 accurately predicted the rank order of butyrate production in high richness communities containing all 5 butyrate producers (Spearman rho = 0.87, $p = 2*10^{-25}$, Supplementary Fig. 9a), while M1 was unable to predict butyrate production (Spearman correlation not significant, $p = 0.68$, Supplementary Fig. 9b). However, M1 outperformed M4 in predicting the rank order of butyrate production for the 3–5 species communities (M1: Spearman rho = 0.84, $p = 9*10^{-43}$, Fig. 2c; M4: Spearman rho = 0.43, $p = 1.4*10^{-8}$, Supplementary Fig. 9c) as well as the high richness communities lacking AC (M1: Spearman rho = 0.5, $p = 1*10^{-6}$, Supplementary Fig. 9b; M4: Spearman rho = 0.21, $p = 0.05$, Supplementary Fig. 9a), indicating the importance of including variation in the composition of functional organisms that matches the variations in the design space of interest. For example, our results indicate that the predictive capability of the model would be improved by training on measurements of communities with all butyrate producers for prediction of communities with all butyrate producers. However, the model's predictive capability would be improved by training on combinations of butyrate producers for prediction of communities containing different sets of butyrate producers.

To further explore experimental designs for building predictive models of microbial communities, we investigated whether the designed community datasets that spanned a wider range of butyrate concentrations would provide more information than randomly selected community datasets for accurate prediction of butyrate production. To test this hypothesis, we trained two

interaction coefficients in the M3 growth model to those from a previous study that developed a gLV model to predict a 12-member subset of our community (PC, BV, BO, BT, BU, DP, CA, EL, FP, CH, BH, and ER) in a different undefined media[16] and found that 27 parameters with magnitude >0.1 h⁻¹ (OD₆₀₀ species j)⁻¹ shared a sign and only five had opposite sign (Supplementary Fig. 8). The high percentage (84%) of qualitatively consistent interaction coefficients inferred based on measurements in two different environmental contexts provides confidence in using gLV parameter estimates as prior information to forecast system behaviors in new environments. In sum, our modeling framework representing pairwise interactions could accurately predict community composition and butyrate production in low and high richness communities and could be used to decipher key microbial interactions impacting metabolic outputs.

The predictive capability of the model of butyrate production in high richness communities required information from other high richness communities, consistent with theoretical work suggesting predictive models of microbiome assembly could be constructed with a fewer number of measurements if the training

additional models including M5, which was trained on the data from M2 plus 80% of the measurements of random 5BP communities and M6, which was trained on the data from M2 plus 80% of the measurements of designed 5BP communities (Supplementary Data 2). We used these models to predict the same validation set consisting of the remaining 20% of high richness communities from both the random and designed sets. Model M6 outperformed model M5 (M5: Spearman rho = 0.75, $p = 3*10^{-7}$, Supplementary Fig. 10a; M6: Spearman rho = 0.81, $p = 5*10^{-9}$, Supplementary Fig. 10b), although both models predicted the rank order of butyrate production with high accuracy. These results demonstrate an advantage of our DTL approach for developing predictive models by using intermediate models to design informative datasets.

**Environmental pH and resource competition impact butyrate production.** Guided by our model and experimental data, we sought to investigate ecological and molecular factors driving butyrate production beyond $H_2S$. We first hypothesized that the low butyrate productivity of specific communities could stem from a global reduction in metabolic activities for the conversion of sugars to fermentation end products. However, the amount of total carbon in acetate, lactate, and succinate was inversely proportional to the amount of carbon in butyrate in high richness communities (Supplementary Fig. 11a), indicating that metabolic tradeoffs dictated the production of specific fermentation end products rather than an overall decrease in metabolic activity. In addition, communities with low butyrate production displayed high acetate concentrations (Supplementary Fig. 11b), suggesting that acetate was not the limiting substrate for butyrate production in high richness communities. These results differ from previous studies that showed acetate was limiting for butyrate production in low richness communities[56,57], suggesting that the mechanisms influencing butyrate production could vary depending on environmental factors such as species richness.

We next sought guidance from specific model parameters for insights into ecological and molecular mechanisms of butyrate production. The regression model M3 identified strong positive production interactions towards AC from *Clostridium asparagiforme* (CG), *Eggerthella lenta* (EL), *Dorea formicigenerans* (DF), and *Collinsella aerofaciens* (CA) that enabled quantitatively accurate predictions of butyrate production (Supplementary Data 3, Model M3). The identification of these interactions upon inclusion of high richness community observations in our model was likely due to increased number of community measurements including each of these species and AC. In the 2–5 species communities, there were only 2–10 examples of communities containing each of these species and AC, whereas in the high richness communities used to train Model M3, there were 28–36 examples containing each species and AC. Due to the improved statistical power of observing pairwise interactions, this result further motivates the use of higher complexity communities to efficiently decipher key inter-species interactions impacting community functions.

Three of these species (EL, CA, and DF) were positively correlated with pH in the high richness communities and EL uniquely increased the pH in monoculture (Supplementary Fig. 12a). Therefore, we directed our attention to environmental pH, which has a major impact on fermentation end product formation by gut microbiota[58–61]. In our experiments, the correlation between butyrate concentration and pH was stronger in 5BP communities (Spearman rho = 0.73, $p = 1^*10^{-57}$) than in 4BP communities (Spearman rho = 0.29, $p = 1^*10^{-4}$) (Supplementary Fig. 12b). Additionally, previous work found that supplemented lactate was converted entirely to butyrate,

propionate, and acetate at pH 5.9 and 6.4, but not at pH 5.2 in batch cultures of fecal inocula[60]. This abrupt metabolic shift at low environmental pH was attributed to inhibition of lactate consumption by AC and *E. hallii* (a closely related lactate-consuming butyrate producer in the clostridial cluster XIVa). These observations combined with the fact that lactate-utilizing butyrate producers, including AC, have been shown to prefer glucose over lactate and produce ~5x more butyrate per unit biomass when grown on lactate versus glucose[52], led us to develop the following mechanistic hypothesis: AC would switch between low butyrate (transformation of sugars into butyrate) and high butyrate (transformation of lactate into butyrate) producing states depending on the environmental pH and availability of key resources (Fig. 4b).

To test the proposed hypothesis, we grew the AC monoculture in media supplemented with acetate and with a broad range of sugar concentrations to represent the varying degrees of resource competition and initial environmental pH values. Consistent with our hypothesis, the butyrate production per unit biomass was inversely related to the sugar concentration (i.e. increasing strength of resource competition) and increased with the initial environmental pH (Fig. 4c, Supplementary Fig. 13). Based on these data, in communities containing pH buffering species such as EL that maintain the pH above a threshold, the butyrate production per unit biomass is dependent on the strength of competition for limited pools of sugars, which support high growth rates and low butyrate production per unit biomass. After sugars are depleted, AC uses an alternative metabolic pathway that transforms lactate into butyrate and enables low growth rates and high butyrate production per unit biomass. The timing of this metabolic shift depends on the rate of sugar consumption and thus the strength of resource competition. In low pH environments, transformation of lactate to butyrate is inhibited and thus AC competes for limited sugars, resulting in butyrate production that is proportional to growth (i.e. no production interactions).

In sum, the proposed mechanism indicates that in a pH-buffered community, resource competition over energy-rich nutrients (i.e. sugars) could enhance butyrate production by AC by triggering a shift in the activity of metabolic pathways from a low to high butyrate-producing state. Environmental pH modification and strong resource competition are more likely to simultaneously occur in high richness communities, which may explain why models trained only on lower-richness communities (M1 and M2 models) were missing the positive production interactions attributed to the combination of these mechanisms. This analysis highlights that our interpretable modeling framework can provide key biological insights into ecological and molecular mechanisms driving community metabolic functions and can illuminate alternative metabolic states by separately quantifying growth and production interactions.

**Biodiversity constrains the achievable range of butyrate production.** The relationship between species richness and ecosystem functions in randomly sampled microbial communities has frequently approximated a saturating function[62]. However, the shape of this function depends on contributions of each species to the given ecosystem function and microbial interactions[63]. To provide insight into the design of communities that display an ecosystem function within desired specifications, we sought to understand the relationship between species richness and the distributions of butyrate and total biomass production by simulating all 33,554,431 possible sub-communities of our 25-species system (Fig. 5). The total community biomass represents a broad function to which all species can contribute and butyrate

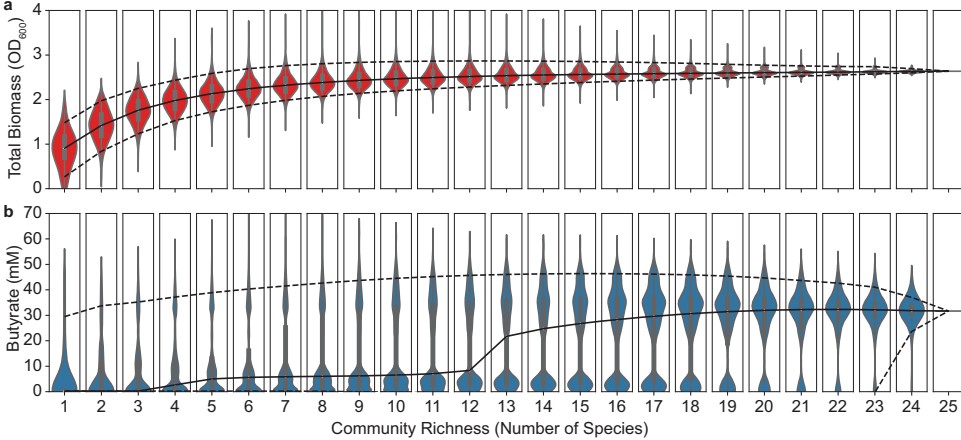

**Fig. 5 Predicted biodiversity–ecosystem function relationships for total biomass and butyrate production.** Violin plots showing the distribution of predicted (**a**) total biomass or (**b**) butyrate concentration for all possible communities at each richness level. Solid lines indicate the median and dashed lines indicate the 5th and 95th percentiles of each distribution (*n* at each community richness value is equal to 25 choose *k*, where *k* is the richness).

production represents a specialized function to which only the five butyrate producers can directly contribute.

The median predicted total biomass is a saturating function of community richness and the upper limits of the distributions peak in the 8–12 species range (Fig. 5a). By contrast, the predicted butyrate concentrations exhibit multimodal distributions at moderate richness and shift to a unimodal distribution at high species richness centered around the 25-species community butyrate concentration (Fig. 5b). For both functions, the median total biomass or butyrate increases as a function of richness indicating that there is a higher probability that a community chosen at random from high richness communities will exhibit high butyrate production. However, the 95th percentiles of the distributions peak at 15 species for butyrate and 12 species for total biomass and display a decreasing trend with richness beyond these thresholds. In sum, increasing species richness increases butyrate and total biomass production on average, while constraining the maximum possible values above a certain richness limit. These results have implications for the design of communities with optimized functional properties by highlighting an optimal range in species richness for identifying communities whose functions deviate from the average behavior.

## Discussion

We demonstrated that community-level functions can be designed using a modeling framework that predicts community assembly and then maps community composition to butyrate production. Our results showed that manipulating the presence/absence of species that do not directly contribute to butyrate production can cause butyrate concentration to vary over a wide range (0–20 or 10–60 mM butyrate in the absence and presence of AC, respectively), highlighting the critical role of microbial interactions on a specialized metabolic function. Benchmarking our model's butyrate prediction accuracy (Fig. 4a, Pearson $r =$ 0.86) against similar DTL approaches for optimizing metabolite production by engineering individual microbes[29,30] (Pearson $r =$ 0.48–0.78) demonstrates that our model is quite accurate. We use our model to show that while ecosystem functions such as butyrate and total biomass production generally increase and saturate as a function of community richness as previously observed[62], richness above a threshold can constrain the minimum and maximum achievable functional outputs of the system. Such constraints may limit the broad efficacy of interventions that attempt to alter microbiome functions and increase diversity

without expanding the set of available organisms, such as autologous FMT[64].

Our results demonstrated that accurate prediction of community function in high richness communities (i.e. >10 species) required measurements of communities at similar richness. Since our model captured only pairwise interactions, a potential explanation of the improved performance of models trained on similar community richness is due to an increased number of observations of pairs of species, which in turn increases the statistical likelihood of accurately inferring a given pairwise interaction. However, an alternative interpretation of the importance of high richness community observations is that our model is approximating communities shaped by higher-order interactions with combinations of pairwise interaction terms. Corroborating the presence of higher-order interactions, butyrate production by AC was shaped by a combinatorial impact of environmental pH modification and competition over sugars. Therefore, the predictive capability of computational models of microbial communities could be improved by training on communities that span a range of species richness levels, rather than implementing a standard procedure of characterizing pairwise communities[16,18,49]. Consistent with this proposed experimental design approach, recent theoretical work has demonstrated a similar perspective[43]. A key difference between low and high richness communities is that constituent members of high richness communities compete more strongly for limited resources and exploit a higher fraction of the available ecological niches, which can alter the metabolic pathway activities in certain species. Further, the combination of strong resource competition with other mechanisms such as pH modification may only occur in communities above a richness threshold such that these combinatorial effects can shape ecological networks differently as a function of species richness. While further work is required to identify whether the difference in our model predictions upon inclusion of high richness community observations is due to higher-order interactions or simply improved statistical inference, it is promising that our pairwise interaction modeling approach was nevertheless able to guide an ecological and mechanistic understanding of butyrate production.

A major strategy for microbiome modulation involves administration of non-resident species predicted to perform a target beneficial function[65], including butyrate producing bacteria[41]. Due to the plasticity of microbial metabolism in response to environmental stimuli, our results demonstrate that it is important to consider both how the resident community will

enable growth of therapeutic bacteria as well as promote the desired metabolic states. Indeed, our results showed that the abundance of butyrate producers may not correlate with butyrate production in the presence of AC due to production interactions that influence the activity of metabolic pathways (Fig. 3c). Notably, we identified high richness communities with higher butyrate production than the five butyrate producers alone (Fig. 3c). These results suggest that the effectiveness of defined bacterial therapies that aim to enhance butyrate production could be improved by manipulating the composition of non-butyrate producers in the microbiome, either through addition of butyrate-enhancing species (i.e. pH-buffering organisms or competitors that promote high butyrate activity states) or microbiome engineering strategies for targeted depletion of butyrate-inhibiting species (e.g. sulfur-reducing bacteria).

While our approach lacks a host-interaction component, the mechanistic nature of insights derived from our model provide generalizable insight into understanding community-level functions in the mammalian gut environment. For instance, DP has been previously associated with IBD[66], attributed to its $H_2S$ activity that inhibits the oxidation of short chain fatty acids by the host via short-chain acyl-CoA dehydrogenase[67]. As highlighted by our results, an additional mechanism through which $H_2S$ producers could contribute to IBD is by inhibiting bacterial production of the anti-inflammatory metabolite butyrate. In a previous study, cecal contents of gnotobiotic mice colonized with an 8-member community plus DP contained less propionate and elevated 3-hydroxybutyrate (upstream intermediate of butyrate production) compared to the DP deletion community (8-member community alone). In this study, the butyrate concentration did not vary between conditions, which could have been masked by host butyrate consumption as the concentration was very low for both communities (<1 mM)[68]. These observations could be explained by $H_2S$ inhibition of bacterial production of butyrate and propionate, which share similar metabolic pathways, consistent with the accumulation of 3-hydroxybutyrate in the former case and decreased propionate production in the latter. In a separate study, a chemical inhibitor of $H_2S$ production resulted in enhanced butyrate production in an anaerobic digestor community, corroborating our results that $H_2S$ can inhibit butyrate production across a broad range of communities[69]. These mechanistic insights could explain associations between inflammatory diseases and other sulfur-reducing bacteria, such as *Bilophila wadsworthia*[70], which is associated with reduced expression of microbial butyrate synthesis pathways in a mouse model of colitis[71]. At the molecular level, $H_2S$ has been observed to form CoA-persulfide through a reaction with coenzyme A, an essential cofactor for SCFA synthesis[72]. Also, $H_2S$ has recently been shown to modulate bacterial metabolism in the gut microbiome via post-translational modification of a wide range of enzymes[73]. These observations combined with our results provide a strong foundation for future work to understand the molecular basis by which $H_2S$ influences bacterial SCFA production and the resulting implications on the host physiology and disease states.

The gut microbiome is exposed to a wide range of dietary substrates and the temporal changes in resource availability can dramatically shape community composition and metabolic activities[74]. In this work, we constructed models of community dynamics and function in a single media under a batch growth period without considering additional physiologically relevant media factors such as dietary fibers and host-secreted factors. While the primary resources in our media comprise simple monomeric compounds rather than some complex substrates found in dietary components (i.e. sugars and amino acids rather than fiber and proteins/peptides), our understanding using this media environment provides a strong foundation which can be extended to understand the impacts of dietary variation on community function. Future work will extend our approach to build model-guided in vitro experimental pipelines for predicting microbiome function in vivo by incorporating these additional physiologically relevant factors as design variables to enable simultaneous exploration of the community and media factor design spaces. Such a model could be used to identify communities with functional responses that are robust to changes in the chemical environment, which may be good candidates for consistent function in vivo. Additionally, host–microbe interactions could be studied in a controlled manner by using co-cultures of synthetic microbiomes with host tissues[75].

While we focused on a predicting a single function (butyrate production), many microbial metabolites directly impact host physiology, with potential combinatorial effects[76]. Thus, designing microbiomes to elicit desired systemic responses from the host may require multifunctional bacterial therapeutic design. Our approach can be extended to design multiple functions by using a multi-output regression as the function model[77]. Alternatively, model interpretability could be sacrificed for potential increases in predictive ability by leveraging flexible machine learning[78]. An additional consideration for such design is that the underlying metabolic networks may drive tradeoffs between production of different molecules, thus constraining the design space. Such relationships could be quantified using flux variability analysis[79] of genome-scale metabolic models[80], providing potentially useful quantitative constraints for multifunctional synthetic microbiome design.

In sum, our methods provide a flexible foundation to explore design strategies for building microbial communities with target functions from the bottom-up and to understand molecular and ecological mechanisms influencing community-level functions. Such methods will advance the design of synthetic microbiomes with predictable functions for deployment into real-world environments. Beyond understanding and manipulating the human gut microbiome to benefit human health, our approach could be used to harness microbiome properties to address numerous challenges facing society. For example, designing microbiomes for efficient nitrogen fixation to reduce the need for fertilizers, degradation of contaminants for detoxification of wastewater and environmental cleanup, and the production of high value compounds from renewable resources for sustainable bioprocessing applications.

## Methods

**Strain maintenance and culturing**. All anaerobic culturing was carried out in an anaerobic chamber with an atmosphere of 2.5 ± 0.5% $H_2$, 15 ± 1% $CO_2$ and balance $N_2$. All prepared media and materials were placed in the chamber at least overnight before use to equilibrate with the chamber atmosphere. The strains used in this work were obtained from the sources listed in Supplementary Data 1 and permanent stocks of each were stored in 25% glycerol at −80 °C. Batches of single-use glycerol stocks were produced for each strain by first growing a culture from the permanent stock in anaerobic basal broth (ABB) media (HiMedia or Oxoid) to stationary phase, mixing the culture in an equal volume of 50% glycerol, and aliquoting 400 μL into Matrix Tubes (ThermoFisher) for storage at −80 °C. Quality control for each batch of single-use glycerol stocks included (1) plating a sample of the aliquoted mixture onto LB media (Sigma-Aldrich) for incubation at 37 °C in ambient air to detect aerobic contaminants and (2) Illumina sequencing of 16S rDNA isolated from pellets of the aliquoted mixture to verify the identity of the organism. For each experiment, precultures of each species were prepared by thawing a single-use glycerol stock and combining the inoculation volume and media listed in Supplementary Data 1 to a total volume of 5 mL (multiple tubes inoculated if more preculture volume needed) for stationary incubation at 37 °C for the preculture incubation time listed in Supplementary Data 1. All experiments were performed in a chemically defined medium (DM38), the composition of which is provided in Supplementary Data 4. This medium was developed by beginning with a previously studied medium that enabled the growth of a subset of our species, replacing some undefined components with defined alternatives (e.g. replacing casamino acids with a defined mixture of purified amino acids), and iteratively adding components to support the growth of diverse gut bacterial

species. Our final medium supported the individual growth of all organisms except *Faecalibacterium prausnitzii*.

**Monoculture dynamic growth quantification**. Each species' preculture was diluted to an $OD_{600}$ of 0.0066 (Tecan F200 Plate Reader, 200 µL in 96-Well Microplate) in DM38 and aliquoted into three replicates of 1 mL each in a 96 Deep Well (96DW) plate and covered with a semi-permeable membrane (Diversified Biotech) and incubated at 37 °C without shaking. At each time point, samples were mixed and $OD_{600}$ was measured by diluting an aliquot of each sample into phosphate-buffered saline (PBS) to be within the linear range of the plate reader.

**Community culturing experiments and sample collection**. To produce all desired community combinations, each species' preculture was diluted to an $OD_{600}$ of 0.0066 in DM38. Community combinations were arrayed in 96DW plates by pipetting equal volumes of each species' diluted preculture into the appropriate wells using a Tecan Evo Liquid Handling Robot inside an anaerobic chamber. Each 96DW plate was covered with a semi-permeable membrane and incubated at 37 °C. After 48 h, 96DW plates were removed from the incubator and samples were mixed. Cell density was measured by pipetting 200 µL of each sample into one microplate and diluting 20 µL of each sample into 180 µL of PBS in another microplate and measuring the $OD_{600}$ of both plates (Tecan F200 Plate Reader). We selected the value that was within the linear range of the instrument for each sample. 200 µL of each sample was transferred to a new 96DW plate and pelleted by centrifugation at 2400×$g$ for 10 min. A supernatant volume of 180 µL was removed from each sample and transferred to a 96-well microplate for storage at −20 °C and subsequent metabolite quantification by high performance liquid chromatography (HPLC). Cell pellets were stored at −80 °C for subsequent genomic DNA extraction and 16S rDNA library preparation for Illumina sequencing. In some experiments, 20 µL of each supernatant was used to quantify pH using a phenol Red assay[81]. Phenol red solution was diluted to 0.005% weight per volume in 0.9% w/v NaCl. Bacterial supernatant (20 µL) was added to 180 µL of phenol red solution, and absorbance was measured at 560 nm (Tecan Spark Plate Reader). A standard curve was produced by fitting the Henderson–Hasselbach equation to fresh media with a pH ranging between 3 and 11 measured using a standard electro-chemical pH probe (Mettler-Toledo). We used Eq. (1) to map the pH values to the absorbance measurements.

$$\mathrm{pH} = \mathrm{p}K_a + b \cdot \log_{10}\left(\frac{A - A_{\min}}{A_{\max} - A}\right) \quad (1)$$

The parameters $b$ and $\mathrm{p}K_a$ were determined using a linear regression between pH and the log term for the standards in the linear range of absorbance (pH between 5.2 and 11) with $A_{\max}$ representing the absorbance of the pH 11 standard, $A_{\min}$ denoting the absorbance of the pH 3 standard and $A$ representing the absorbance of each condition.

**Sulfide titration experiment**. Sodium sulfide stock solution and standards were prepared in nitrogen purged water and 1% zinc acetate, respectively, and stored in stoppered, capped Hungate tubes at maximum fill volume to limit volatilizing and oxidizing of sulfide during setup. Each species' preculture was diluted to an $OD_{600}$ of 0.0066 in DM38. Four falcon tubes of 1.11× concentrated DM38 were aliquoted, and brought to target sulfide concentration by adding an appropriate volume of sodium sulfide stock solution and a balance of sodium chloride solution, of equimolar sodium content, to a final media volume of 50 mL and DM38 of 1×. This procedure was performed one sulfide concentration at a time, followed by aliquoting into 1.6 mL microfuge tubes, inoculation of each strain, and capping to minimize sulfide volatizing during setup. FP cultures were supplemented with 1 g/L bacto yeast extract (BD) and 33 mM sodium acetate (Sigma Aldrich) prior to capping microfuge tubes. Capped microfuge tubes were incubated at 37 °C for 24 h at which point the $OD_{600}$ was measured (Tecan F200 Plate Reader, 200 µL in 96-Well Microplate) and supernatants were collected for organic acid quantification via HPLC. Initial sulfide concentrations were measured in blank microfuge tube aliquots of each sulfide concentration via the Cline assay[82], to account for degradation of the sulfide stock during experimental setup. Briefly, 14.8 µL of Cline reagent was added to 185.2 µL of culture supernatant and incubated in a sealed 96-Well Microplate for 2 h before diluting in 1% zinc acetate (Fisher) to the linear range of absorbance measurement at 667 nm (Tecan Spark Plate Reader). A standard curve was prepared similarly using sodium sulfide fixed in 1% zinc acetate. The Cline reagent was prepared by dissolving 1.6 g N,N-dimethyl-*p*-phenylenediamine sulfate (Acros Organics) and 2.4 g FeCl₃ (Fisher) in 100 mL 50% v/v HCl (Fisher) in water.

**pH and sugars cross-titration experiment**. In the pH and sugar titration experiment, the media DM38 was supplemented with 33 mM sodium acetate (Sigma) and excluded glucose, arabinose, and maltose. This media was split into four aliquots, each of which was pH adjusted to the specified pH using 5 M HCl. Each aliquot was then divided in half and glucose, arabinose, and maltose were added to one half to achieve the same concentrations as in DM38. For each pH, the media with and without sugars were serially diluted in ratios necessary to achieve the indicated fraction of sugars. AC was cultured in these media for 48 h, after

which samples were collected and measured as described for the community culturing experiments.

**HPLC quantification of organic acids**. Supernatant samples were thawed in a room temperature water bath before addition of 2 µL of $H_2SO_4$ to precipitate any components that might be incompatible with the running buffer. The samples were then centrifuged at 2400 × $g$ for 10 min and then 150 µL of each sample was filtered through a 0.2 µm filter using a vacuum manifold before transferring 70 µL of each sample to an HPLC vial. HPLC analysis was performed using either a Thermo-Fisher (Waltham, MA) Ultimate 3000 UHPLC system equipped with a UV detector (210 nm) or a Shimadzu HPLC system equipped with a SPD-20AV UV detector (210 nm). Compounds were separated on a 250 × 4.6 mm Rezex© ROA-Organic acid LC column (Phenomenex Torrance, CA) run with a flow rate of 0.2 mL min⁻¹ and at a column temperature of 50 °C. The samples were held at 4 °C prior to injection. Separation was isocratic with a mobile phase of HPLC grade water acidified with 0.015 N $H_2SO_4$ (415 µL L⁻¹). At least two standard sets were run along with each sample set. Standards were 100, 20, and 4 mM concentrations of butyrate, succinate, lactate, and acetate, respectively. For most runs, the injection volume for both sample and standard was 25 µL. The resultant data was analyzed using the Thermofisher Chromeleon 7 software package or Shimadzu LabSolutions software package.

**Genomic DNA extraction and sequencing library preparation**. Genomic DNA was extracted from cell pellets using a modified version of the Qiagen DNeasy Blood and Tissue Kit protocol. First, pellets in 96DW plates were removed from −80 °C and thawed in a room temperature water bath. Each pellet was resuspended in 180 µL of enzymatic lysis buffer (20 mM Tris–HCl (Invitrogen), 2 mM sodium EDTA (Sigma-Aldrich), 1.2% Triton X-100 (Sigma-Aldrich), 20 mg/mL lysozyme from chicken egg white (Sigma-Aldrich)). Plates were then covered with a foil seal and incubated at 37 °C for 30 min with orbital shaking at 600 RPM. Then, 25 µL of 20 mg mL⁻¹ Proteinase K (VWR) and 200 µL of Buffer AL (QIAGEN) were added to each sample before mixing with a pipette. Plates were then covered by a foil seal and incubated at 56 °C for 30 min with orbital shaking at 600 RPM. Next, 200 µL of 100% ethanol (Koptec) was added to each sample before mixing and samples were transferred to a nucleic acid binding (NAB) plate (Pall) on a vacuum manifold with a 96DW collection plate. Each well in the NAB plate was then washed once with 500 µL buffer AW1 (QIAGEN) and once with 500 µL of buffer AW2 (QIAGEN). A vacuum was applied to the Pall NAB plate for an additional 10 min to remove any excess ethanol. Samples were then eluted into a clean 96DW plate from each well using 110 µL of buffer AE (QIAGEN) preheated to 56 °C. Genomic DNA samples were stored at −20 °C until further processing.

Genomic DNA concentrations were measured using a SYBR Green fluorescence assay and then normalized to a concentration of 1 ng µL⁻¹ by diluting in molecular grade water using a Tecan Evo Liquid Handling Robot. First, genomic DNA samples were removed from −20 °C and thawed in a room temperature water bath. Then, 1 µL of each sample was combined with 95 µL of SYBR Green (Invitrogen) diluted by a factor of 100 in TE buffer (Integrated DNA Technologies) in a black 384-well microplate. This process was repeated with two replicates of each DNA standard with concentrations of 0, 0.5, 1, 2, 4, and 6 ng µL⁻¹. Each sample was then measured for fluorescence with an excitation/emission of 485/535 nm using a Tecan Spark plate reader. Concentrations of each sample were calculated using the standard curve and a custom Python script was used to compute the dilution factors and write a worklist for the Tecan Evo Liquid Handling Robot to normalize each sample to 1 ng µL⁻¹ in molecular grade water. Samples with DNA concentration <1 ng µL⁻¹ were not diluted. Diluted genomic DNA samples were stored at −20 °C until further processing.

Amplicon libraries were generated from diluted genomic DNA samples by PCR amplification of the V3–V4 of the 16S rRNA gene using custom dual-indexed primers (Supplementary Data 5) for multiplexed next-generation amplicon sequencing on Illumina platforms (Method adapted from Venturelli et al. *Mol. Syst. Biol.*, 2018). Primers were arrayed in skirted 96-well PCR plates (VWR) using an acoustic liquid handling robot (Labcyte Echo 550) such that each well received a different combination of one forward and one reverse primer (0.1 µL of each). After liquid evaporated, dry primers were stored at −20 °C. Primers were resuspended in 15 µL PCR master mix (0.2 µL Phusion High Fidelity DNA Polymerase (Thermo Scientific), 0.4 µL 10 mM dNTP solution (New England Biolabs), 4 µL 5× phusion HF buffer (Thermo Scientific), 4 µL 5 M Betaine (Sigma-Aldrich), 6.4 µL Water) and 5 µL of normalized genomic DNA to give a final concentration of 0.05 µM of each primer. Primer plates were sealed with Microplate B seals (Bio-Rad) and PCR was performed using a Bio-Rad C1000 Thermal Cycler with the following program: initial denaturation at 98 °C (30 s); 25 cycles of denaturation at 98 °C (10 s), annealing at 60 °C (30 s), extension at 72 °C (60 s); and final extension at 72 °C (10 min). 2 µL of PCR products from each well were pooled and purified using the DNA Clean & Concentrator (Zymo) and eluted in water. The resulting libraries were sequenced on an Illumina MiSeq using a MiSeq Reagent Kit v3 (600-cycle) to generate 2 × 300 paired-end reads.

**Bioinformatic analysis for quantification of species abundance**. Sequencing data were demultiplexed using Basespace Sequencing Hub's FastQ Generation

program. Custom python scripts were used for further data processing (method adapted from Venturelli et al. *Mol. Syst. Biol.*, 2018)[16]. Paired end reads were merged using PEAR (v0.9.10)[83] after which reads without forward and reverse annealing regions were filtered out. A reference database of the V3–V5 16S rRNA gene sequences was created using consensus sequences from next-generation sequencing data or Sanger sequencing data of monospecies cultures. Sequences were mapped to the reference database using the mothur (v1.40.5)[84] command classify.seqs (Wang method with a bootstrap cutoff value of 60). Relative abundance was calculated as the read count mapped to each species divided by the total number of reads for each condition. Absolute abundance of each species was calculated by multiplying the relative abundance by the $OD_{600}$ measurement for each sample. Samples were excluded from further analysis if >1% of the reads were assigned to a species not expected to be in the community (indicating contamination) or if they had <1000 total reads and $OD_{600} > 0.1$ (indicating that there were insufficient reads for analysis and this was not due to lack of community growth).

### gLV models and training.

We used a model with two modules: the gLV model to predict composition of the assembled community and a regression model with interaction terms to predict butyrate production as a function of the predicted community composition (Fig. 1a). The gLV model is a set of $N$ coupled first-order ordinary differential equations, where $N$ denotes the number of species, shown in Eq. (2).

$$\frac{1}{X_i}\frac{dX_i}{dt} = r_i + \sum_{j=1}^{N} a_{ij}X_j \quad (2)$$

The species $X_i$ is the abundance of species $i$, $r_i$ is a parameter that represents the basal growth rate of species $i$, and $a_{ij}$ is a parameter that represents interactions by modifying the growth rate of species $i$ proportional to the abundance of species $j$. To prevent unbounded growth, $a_{ij}$ is constrained to be negative when $i = j$, representing intra-species competition. This model has previously been used to understand and predict the behavior of microbial communities[16] and provides an interpretable model form (e.g. which interspecies interactions are important) without introducing an excessive number of parameters (e.g. mechanistic models[85]).

We used a Bayesian parameter inference approach to estimate parameters for the gLV model from experimental measurements (adapted from Shin et al., *PLoS Comput. Biol.*, 2019[44]). Briefly, our method has a prior distribution for each model parameter and then varies the parameters to fit the model to the measured species abundances (mean of biological replicates) while penalizing deviations from the parameter prior distributions. These penalties provide a regularization effect, which is necessary when the model is underdetermined. We used L2 regularization because we expected inter-species competition to be prevalent and thus did not expect many interaction parameters to be negligible. After an optimal parameter set is found, this process is repeated hundreds of times after applying random noise to the experimental data proportional to the measured experimental variance to generate an ensemble of parameter sets (i.e. the posterior distribution). This posterior distribution is then used as the prior distribution when updating the model with new data. We adapted a previous implementation of this method in Julia for this work.

Before training the model on any data, we assumed a normally distributed prior for each parameter with mean of 0 and standard deviation equal to 1. We then trained the gLV model on time-series measurements of monoculture growth for each species, estimating a posterior distribution for each $r_i$ and $a_{ii}$ parameter (other $a_{ij}$ posterior distributions were equal to the prior distribution). We used this posterior distribution as a prior distribution to update the model with the pairwise community data and generated the gLV module of Model M1, where posterior distributions were estimated from experimental data for $r_i$, $a_{ii}$, and $a_{ij}$ where species $i$ and species $j$ co-occurred in the experimental data and the posterior distribution of $a_{ij}$ for unobserved pairs was equal to the prior. We similarly updated the model using the 3–5-member community experiments to generate Model M2. Regularization coefficients for each iteration of the model updating process are shown in Supplementary Data 6.

The gLV modules of Models M1 and M2 were underdetermined due to pairs of species never being observed in the same community within the training dataset. To generate parameters for these unobserved interactions, we used a matrix imputation approach to estimate the interaction parameters informed by the phylogenetic relatedness of species. First, we sorted the $a_{ij}$ interaction parameter matrix such that the rows and columns occurred in the same order as the phylogenetic tree (Fig. 1c). Next, we used $K$-nearest neighbors matrix imputation with $K = 2$ to estimate interaction parameters for species that were not observed in the training data (implemented in Python 3 using the fancyimpute package, https://pypi.org/project/fancyimpute/). This process was repeated independently for each parameter set in the posterior distribution.

While the parameter optimization portion of this model-training process had previously been found to scale with increasing number of pairwise community datasets[44], we found that the optimization problem became intractable when attempting to estimate parameters from high richness community data (i.e. >10 species), due to infeasible memory requirements resulting in an inability

to converge on a solution. To address this problem, we used the nonlinear programming solver FMINCON in MATLAB to generate the gLV module of Model M3 by training on all data simultaneously. While this approach is much less computationally efficient than the previous approach (taking hours rather than minutes to converge for datasets containing only low richness communities), it does not have the same memory-related convergence issues for datasets containing high richness communities and will find a solution given sufficient time. Using this method, the cost function for the optimization algorithm is computed using an ODE solver to simulate each community and the sum of mean squared errors for the community is computed and added to a L2 regularization term penalizing the magnitude of the parameter vector. To ensure that the model did not sacrifice the goodness of fit to the time-series monospecies data, the mean squared errors for these data were weighted more highly. The resulting optimization function was Eq. (3).

$$\varphi = \sum_{k \in \text{Single}} (X_{\exp,k} - X_{\text{model},k})^2 + w \sum_{l \in \text{Dynamic}} (X_{\exp,l} - X_{\text{model},l})^2 + \lambda \sum_{j \in \text{Params}} \theta_j^2 \quad (3)$$

In this equation, single denotes the set of experiments where only the end point community composition was measured, dynamic indicates the set of time-series monospecies measurements, $w$ is the weighting factor for the time-series monospecies measurements, and $\lambda$ represents the regularization coefficient. The FMINCON function identifies a parameter estimate which minimizes the cost function. We provided the median parameter values from Model M2 as an initial guess for the FMINCON function. We repeated this process with various values of $\lambda$ and $w$ to find a parameter set that simultaneously fits the Dynamic and Single datasets with maximal regularization penalty to prevent overfitting to the data (Supplementary Data 6). We used a procedure based on the one described above for the Julia implementation to generate an ensemble of parameter sets (i.e. the posterior distribution) using FMINCON. Because each iteration of the FMINCON parameter estimation took several hours to complete, we massively parallelized the generation of each of the hundreds of parameter sets in the ensemble using resources from the UW-Madison Center for High Throughput Computing.

### Regression models and training.

We used a regression model with interaction terms to represent microbial community metabolite release. We chose this model based on the following derivation beginning with typical models used for understanding metabolite release in bioprocess engineering[86]. First, we considered an Eq. (4) determining productivity as a function of biomass concentration,

$$\frac{dP}{dt} = \sum_{i \in \text{BP}} q_{P,i}X_i \quad (4)$$

where $P$ represents the product concentration (mM), $X_i$ denotes the biomass concentration of species $i$, and $q_{P,i}$ represents the specific productivity of species $i$ (mM $OD_{600}^{-1}$ $h^{-1}$). If we assume that the specific productivity is proportional to the specific growth rate (i.e. a growth coupled metabolite release, as would be expected for fermentation products which are a byproduct of growth), we can define it as Eq. (5).

$$q_{P,i} = \rho_i\mu_i = \rho_i\left(r_i + \sum_{j=1}^{N} a_{ij}X_j\right) \quad (5)$$

Here, $\rho_i$ represents the production per unit growth of species $i$ and $\mu_i$ represents the specific growth rate of species $i$, which is then substituted from the gLV model Eq. (2). Further, $\rho_i$ can be a function of the abundances of other species in the system to capture behavior where the presence of one species causes another to alter its metabolism, as in Eq. (6).

$$\rho_i = \rho_{i0} + \sum_{j=1}^{N} \rho_{ij}X_j \quad (6)$$

Combining Eqs. (4)–(6) yields Eq. (7) describing the overall productivity.

$$\frac{dP}{dt} = \sum_{i \in \text{BP}} \left(\rho_{i0} + \sum_{j=1}^{N} \rho_{ij}X_j\right)\left(r_i + \sum_{j=1}^{N} a_{ij}X_j\right)X_i \quad (7)$$

To determine the product concentration at a specific time (e.g. 48 h as in our experiments), we can integrate Eq. (6) in time, yielding Eq. (8).

$$P(T) = \int_0^T \sum_{i \in \text{BP}} \left(\rho_{i0} + \sum_{j=1}^{N} \rho_{ij}X_j\right)\left(r_i + \sum_{j=1}^{N} a_{ij}X_j\right)X_i dt$$
$$= \sum_{i \in \text{BP}} \left[\int_0^T \rho_{i0}r_iX_i dt + \sum_{i=1}^{N}\int_0^T (\rho_{i0}a_{ij} + \rho_{ij}r_i)X_iX_j dt + \sum_{i=1}^{N}\int_0^T \rho_{ij}a_{ij}X_iX_j^2 dt\right] \quad (8)$$

Because these integrals are complex, involving the time integrals of multiple species, and the lack of time resolution of our data, we chose to make linear approximations as described in Eqs. (9) and (10). Additionally, we chose to assume that third-order interactions of the type shown in Eq. (11) were small for two reasons. First, we reasoned that species were unlikely to have large impacts on both the growth (via $a_{ij}$) and metabolic shifts (via $\rho_{ij}$) on the same species, thus their product would be small. Second, including such interactions would introduce 325 additional parameters to our model, reducing our ability to reasonably infer all parameter values and increasing the chances of overfitting. Together, these

assumptions result in the linear regression model with interactions described in Eq. (12).

$$\int_0^T \rho_{i0} r_i X_i \, dt \approx \alpha_{i,0} + \alpha_i X_i(T) \tag{9}$$

$$\int_0^T (\rho_{i0} a_{ij} + \rho_{ij} r_i) X_i X_j \, dt \approx \beta_{ij,0} + \beta_{ij} X_i(T) X_j(T) \tag{10}$$

$$\int_0^T \rho_{ij} a_{ij} X_i X_j^2 \, dt \approx 0 \tag{11}$$

$$P(T) = \sum_{i \in BP} (\alpha_{i,0} + \alpha_i X_i(T)) + \sum_{j \in BP} \sum_{k \in ALL} (\beta_{jk,0} + \beta_{jk} X_j(T) X_k(T)) \tag{12}$$

Note that the summations in Eq. (12) refer only to those species included in a community, so the intercepts $\alpha_{i,0}$ and $\beta_{jk,0}$ only apply when the corresponding species were added to the community. These intercepts allow for production of butyrate to be attributed to the presence of butyrate producers even if their abundance is low at the time under consideration, which could be due to non-monotonic growth dynamics (i.e. early growth and butyrate production followed by inhibition from interactions causing a decrease in abundance). A mathematical consequence of these intercepts is a lower bound on the confidence intervals of butyrate production across communities with a conserved set of butyrate producers, as observed in Fig. 2b. The simplified $\beta_{ij}$ parameters capture additional modifications to productivity via interactions and are thus informative in guiding investigation of which pairs of species may be participating in production interactions. The resulting regression model had 450 total parameters. Model fitting was performed using custom scripts written in MATLAB (M1 and M2, using lasso function) and Python (M3, using scikit-learn[87]). We used L1 regularization to minimize the number of nonzero parameters. Regularization coefficients were chosen by using 10-fold cross validation and choosing the coefficient value with the lowest median mean-squared error for the test data. For models M1 and M2, ensembles of regression models were generated, one for each possible combination of butyrate producers, where samples containing butyrate producers from outside of each set were excluded. In this case, butyrate production from less productive species (e.g. FP) were small compared to more productive species (e.g. AC, ER, RI, CC) thus reducing the model accuracy for communities lacking the high productivity species. For Model M3, one regression model was generated using all data because all communities of interest contained highly productive butyrate producers.

**Model simulations to predict new communities.** Custom MATLAB scripts were used to predict community assembly and butyrate production, for many communities as described in the text (e.g. all communities containing all 5 butyrate producers for Fig. 3a). For each community, the growth dynamics were simulated using each parameter set from the posterior distribution of the gLV model. The resulting community compositions for each simulation were an input to the regression model to predict butyrate concentration. Statistics on the resulting distributions of butyrate concentration and abundance of each species were stored for later plotting. Because of the large number of communities and the large number of parameter sets (i.e. hundreds of simulations per community), we used parallel computing (MATLAB parfor) to complete the simulations in a reasonable timeframe (~4 days for the communities in Fig. 3a). For the communities in Fig. 5, we simulated only with the best fit parameters due to the large number of possible communities (33,554,432 communities).

**Statistics and reproducibility.** The experiments of various communities were performed in nine different batches and the full community was included in each of these batches to quantify variation across the different experimental batches. The 24-species "leave-one-out" communities were also repeated in three of the batches. The variance of these replicated communities can be seen in Figs. 2a, 3c,d, and 4a. The sulfide titration experiment was replicated on three separate occasions with slight differences in experimental approach while optimizing the procedure to limit loss of $H_2S$ to the gas phase and the results agreed at least qualitatively with the data shown on each occasion.

**Reporting summary.** Further information on research design is available in the Nature Research Reporting Summary linked to this article.

## Data availability
Additional data supporting the findings described in this work are available from the corresponding author under reasonable request. The processed data for all community experiments and simulation results are available in a Github Repository: https://github.com/RyanLincolnClark/DesignSyntheticGutMicrobiomeAssemblyFunction[88]. The Illumina Sequencing data are available via Zenodo at https://doi.org/10.5281/zenodo.4642238. Source data are provided with this paper.

## Code availability
Scripts for model training and simulations are available in a Github Repository here: https://github.com/RyanLincolnClark/DesignSyntheticGutMicrobiomeAssemblyFunction[88].

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

## Acknowledgements

We would like to thank Sungho Shin, Jordan Jalving, and Victor Zavala for their advice related to implementing Julia parameter estimation methods. In addition, we are grateful to Mayank Baranwal and Alfred Hero for conversations which inspired the matrix imputation approach for estimating unobserved interaction parameters. We would like to thank Federico Rey for generously taking the time to provide advice that improved the manuscript. Research was sponsored by the National Institutes of Health and was accomplished under Grant Number R35GM124774, the University of Wisconsin-Madison Office of the Chancellor and Vice Chancellor for Research and Graduate Education with funding from the Wisconsin Alumni Research Foundation, National Institute of Biomedical Imaging and Bioengineering under grant number R01EB030340 and the Multi University Research Initiative (MURI) from under grant number W911NF-19-1-0269. S.E.H. was supported by the National Institute of General Medical Sciences of the National Institutes of Health under Award Number T32GM008349. R.L. C. was supported in part by an NHGRI training grant to the Genomic Sciences Training Program (T32 HG002760). This research was performed using the computing resources and assistance of the UW-Madison Center for High Throughput Computing (CHTC) in the Department of Computer Sciences. The CHTC is supported by UW-Madison, the Advanced Computing Initiative, the Wisconsin Alumni Research Foundation, the Wisconsin Institutes for Discovery, and the National Science Foundation, and is an active member of the Open Science Grid, which is supported by the National Science Foundation and the U.S. Department of Energy's Office of Science.

## Author contributions

O.S.V. and R.L.C conceived the study. R.L.C., J.J.H., S.E.H., and B.M.C. carried out the experiments. R.L.C. implemented computational modeling. R.L.C., S.E.H., and O.S.V. analyzed the data. B.M.C. proposed inhibition of butyrate production by hydrogen sulfide. D.A.-N. and D.M.S. designed and implemented metabolite measurements. O.S.V. secured funding. R.L.C. and O.S.V. wrote the paper and all authors provided feedback on the manuscript.

## Competing interests

The authors declare no competing interests.
