## [Peer Review File · Nature Communications]

Reviewers' Comments:

Reviewer #1:

Remarks to the Author:

Design of synthetic human gut microbiome assembly and butyrate production
Clark et. al.

The manuscript by Clark describes a novel approach to the design of microbial communities, which combines a gLV to predict community composition and then a linear model to predict output metabolites given community composition. This builds on an existing approach to optimisation (described in Shin et. al. 2019). The authors focus on the design and prediction of butyrate production, which is of high interest in human and animal health. The authors use this approach to disentangle a number of higher order effects that influence butyrate production within these communities, which importantly can't be determined solely from pairwise interaction knowledge. One important finding is that in low richness communities (ie low number of species) butyrate production is not necessarily correlated with butyrate producer concentration, which highlights the need for design approaches that handle the complex interaction networks. This also demonstrates that species not directly involved in butyrate production are important for community function (and hence manipulation).

Overall, I like the approach taken. What is nice about this Bayesian framework is that it lends itself well to the design-test-learn (DTL) cycle whereby information can be added in an iterative manner. Indeed, the authors pass through a number of iterations of the DTL cycle including information from increasingly rich communities.

The paper itself is quite hard to follow in places and I have a number of points that I would like clarified. I also found some information missing on certain aspects of the work, and have included some more general comments and suggestions for improvement of the manuscript.

COMMENTS

Throughout the manuscript the authors use the terms low and high species richness, without ever defining what these terms are. I found this confusing. Is high richness 11-17 species and low richness 4-5 species. Can the authors be more specific?

Figure 2 in general could be improved for readability and understandability. Fig 2b could be expanded out into more panels. Questions here:

- It isn't clear why there is a lower bound on the posterior predictions. Is this based solely on the amount of butyrate produced by the producer strain? This could be clarified.
- What is happening in AC, RI? It looks like the model has broken down?

I found the main text from line 208 describing results in Figure 3 difficult to follow as the narrative jumps around the subfigures. Can this part of the text be improved, and perhaps figure 3 modified to make it clearer to the reader to follow?

Figure 3a: I didn't understand why the data point for the community of 5 producers (black star) wasn't included here.

Ln150: How was the defined media, DM38, developed? Although there is a list of additives, it wasn't clear how these were chosen. There is a short section on media at the end of the discussion, but how might the choice of this defined media influence the results and findings?

Ln192: The authors state that "the model prediction exhibited good agreement with the rank order of butyrate production". However, from Fig 2b, this doesn't look like the case. Can the authors make another plot where they directly compare their predictions to the data.

Ln710: The authors apply their existing fitting approach in Shin 2019 but then move to a more standard optimisation approach. So it wasn't clear what was gained by using the nlp approach in the first place.

Ln739: In the model, the authors assume that specific productivity is proportional to the specific growth rate. But isn't this violated in the main findings?

TYPOGRAPHICAL

Ln703: Figure 2a -> Figure 1c ?

Figure 2: 3-5 BPB -> 3-5 BP ?

Reviewer #2:

Remarks to the Author:

In this manuscript, Clark et al. build and iteratively train a bottom-up model that is able to predict butyrate production in simplified communities of Human gut bacteria. Using this model, they identify factors that affect butyrate production, both by modulating the abundance of butyrate producers, and their per capita production.

This is a very interesting and well-executed study that provides an impressive example of successfully predicting microbial community function, which is an important and timely subject with important implications for managing natural communities and designing artificial ones. It is especially notable that the authors are able to accurately predict butyrate production in communities composed of as many as 25 species - previous research in the field typically focused on communities composed of ~10 species, and predictability typically deteriorates with increasing richness. I only have a few comments that are mainly about clarity and presentation, as detailed below.

Specific comments

I found it difficult to understand exactly what are the proposed benefits of the Design-Test-Learn (DTL) paradigm in this context, and how it should be utilized. The fact that the models based on more measurement, including previously unmeasured interaction, outperform the initial simple models is not surprising. For example, The DTL paradigm can be beneficial if it can be used to guide the selection of the most informative measurements following an initial set of measurements. I'm sure that was the case in this work. If so, how were future measurements chosen? And were they more informative than an equivalent number of "naive" measurements? It would be useful for the authors to provide general guidance to readers regarding the best way to implement the DTL paradigm when building a predictive model of community function. To clarify, this work is very valuable even if in this case the DTL approach is not more predictive than a model based on "one-shoot" measurements. But if the authors want to highlight this approach as an important aspect of the work they need to better demonstrate its utility, and explain how it was employed.

Relatedly, there is some ambiguity regarding the utility of measurement of multispecies communities vs species pairs. The results section suggests that the utility of high-richness communities measurements is that they enable constructing models using fewer measurements (II. 296-9). I interpreted this statement as saying that species indeed predominantly interact in a pairwise manner, with few higher-order interactions (HOI), but these pairwise interaction strengths can be estimated more efficiently from measurement of high-richness communities rather than all pairwise measurements. Related to the above point regarding the DTL paradigm, it's not clear whether that is true in this study (whether the measured higher-richness

communities are chosen rationally or not), since the total number of measurements used to train the last model (M3) actually exceeds that number required to exhaustively measure all species pairs.

In contrast, the discussion (second paragraph) argues that interactions actually differ in low vs high richness communities, indicating the presence of HOI. In this case, measurements of high richness communities is not a matter of inference efficiency, but rather that the effective strengths of pairwise interactions differ in these communities compared to in low richness communities. It would be valuable to clarify this point and the authors' interpretation of the actual meaning of the inferred pairwise interaction coefficients.

In panel 3f, I don't understand how is the inset consistent with the data in the main panel. In the inset, there is a clear decrease in butyrate production by AC in the presence of S within 2h. In contrast, in the panel there is no difference, or a slight increase.

It was sometimes hard to find the exact number of replicates of different measurements. Adding the number of replicates in the figure captions would help.

Some of the figures are quite small and the details are hard to make out. Especially Fig. 2.

Before reading the Methods section, I was confused regarding the exact meaning and units of F. That is, that it is the butyrate concentration at a particular time point, rather than its rate of production. Perhaps edit line 122 to say "mapping community composition (i.e. abundance of each species) at a specific time point to the concentration of an output metabolite *at that time*".

In the caption of Fig.1 panel (a) is labeled (b).

L665: were samples filtered out if they had OD600 smaller than 0.1, or larger than 0.1?

REVIEWER COMMENTS

Reviewer #1 (Remarks to the Author):

Design of synthetic human gut microbiome assembly and butyrate production
Clark et. al.

The manuscript by Clark describes a novel approach to the design of microbial communities, which combines a gLV to predict community composition and then a linear model to predict output metabolites given community composition. This builds on an existing approach to optimisation (described in Shin et. al. 2019). The authors focus on the design and prediction of butyrate production, which is of high interest in human and animal health. The authors use this approach to disentangle a number of higher order effects that influence butyrate production within these communities, which importantly can't be determined solely from pairwise interaction knowledge.

Once important finding is that in low richness communities (ie low number of species) butyrate production is not necessarily correlated with butyrate producer concentration, which highlights the need for design approaches that handle the complex interaction networks. This also demonstrates that species not directly involved in butyrate production are important for community function (and hence manipulation).

Overall, I like the approach taken. What is nice about this Bayesian framework is that it lends itself well to the design-test-learn (DTL) cycle whereby information can be added in an iterative manner. Indeed, the authors pass through a number of iterations of the DTL cycle including information from increasingly rich communities.

The paper itself is quite hard to follow in places and I have a number of points that I would like clarified. I also found some information missing on certain aspects of the work, and have included some more general comments and suggestions for improvement of the manuscript.

COMMENTS

Throughout the manuscript the authors use the terms low and high species richness, without ever defining what these terms are. I found this confusing. Is high richness 11-17 species and low richness 4-5 species. Can the authors be more specific?

The reviewer is correct in their understanding that "low richness communities" refer to communities with 5 or fewer species and "high richness communities" refers to communities with 11 or more species. We have added some clarifying statements in Lines 175, 213, and 266.

Figure 2 in general could be improved for readability and understandability. Fig 2b could be expanded out into more panels.

We modified Figure 2 by expanding into multiple panels, separating the concepts of experimental design (Figure 2b) from the experimental validation (Figure 2c). We believe that the modified figure with the associated modified descriptions described below clarifies these concepts.

Questions here:

- It isn't clear why there is a lower bound on the posterior predictions. Is this based solely on the amount of butyrate produced by the producer strain? This could be clarified.

The lower bound on the prediction confidence intervals is a consequence of the intercepts in the regression model for predicting butyrate from the gLV-predicted species abundance (i.e. non-zero butyrate production when butyrate producer abundance is zero). We have modified the text in the methods (Lines 864-872) to clarify this point and explain its biological significance.

- What is happening in AC, RI? It looks like the model has broken down?

We thank the reviewer for requesting clarification about the communities containing AC and RI. We have added the following text to the manuscript (Lines 190-203) to describe the predicted behavior of communities containing both AC and RI:

*“In addition, we observed variations in the shapes of the probability distributions based on how the uncertainty in growth prediction propagated through the regression model. For instance, the butyrate concentration in the AC, *Roseburia intestinalis* (RI) pairwise community was lower than the AC monoculture, even though RI was low abundance, resulting in a high magnitude negative parameter in the regression model for a production interaction between AC and RI (**Table S3**). Due to the uncertainty in the growth parameters, the model predicted that RI would grow substantially in a subset of the 3-5 member simulations containing both AC and RI. The variability in predicted RI growth combined with the high magnitude negative interaction parameter between AC and RI resulted in distributions where the median butyrate concentration was high (i.e. for simulations where RI did not grow substantially), and the 60 percent confidence interval extended to 0 mM butyrate (i.e. when RI grew substantially) (**Figure 2b**). In sum, these results demonstrate that the shape of the predicted probability distributions can provide information about the uncertainty in species growth based on experimental observations.”*

I found the main text from line 208 describing results in Figure 3 difficult to follow as the narrative jumps around the subfigures. Can this part of the text be improved, and perhaps figure 3 modified to make it clearer to the reader to follow?

We apologize for the confusion and have modified the text and references to specific figure panels to make this section flow more smoothly.

Figure 3a: I didn't understand why the data point for the community of 5 producers (black star) wasn't included here.

The black star representing the community of 5 butyrate producers is in the upper right-hand corner of Figure 3a.

Ln150: How was the defined media, DM38, developed? Although there is a list of additives, it wasn't clear how these were chosen. There is a short section on media at the end of the discussion, but how might the choice of this defined media influence the results and findings?

We added the following description to the Methods and Discussion sections to address this point:

(Lines 600-605): “This medium was developed by beginning with a previously studied medium that enabled the growth of a subset of our species, replacing some undefined components with defined alternatives (e.g. replacing casamino acids with a defined mixture of purified amino acids), and iteratively adding components that we hypothesized could enable growth of others.

Our final medium supported the individual growth of all organisms except *Faecalibacterium prausnitzii*.”

(Lines 549-553): “While the primary resources in our media comprise simple monomeric compounds rather than some complex substrates found in dietary components (i.e. sugars and amino acids rather than fiber and proteins/peptides), our understanding using this media environment provides a strong foundation which can be extended to understand the impacts of dietary variation on community function.”

Ln192: The authors state that “the model prediction exhibited good agreement with the rank order of butyrate production”. However, from Fig 2b, this doesn’t look like the case. Can the authors make another plot where they directly compare their predictions to the data.

The scatter plot of predicted versus measured butyrate concentration was originally shown as Fig. S3b and these data have been moved to Fig. 2c in the revised manuscript.

Ln710: The authors apply their existing fitting approach in Shin 2019 but then move to a more standard optimisation approach. So it wasn’t clear what was gained by using the nlp approach in the first place.

We found that while the approach adapted from Shin et al. was more computationally efficient for training on datasets containing only low richness communities (taking only a few minutes to converge compared to hours for the MATLAB FMINCON approach for a single dataset), issues related to infeasible memory requirements precluded convergence for datasets containing high richness communities. Thus, we used the Shin et al. approach for datasets containing only low richness communities, where it was more efficient, and the MATLAB FMINCON approach for the latter datasets, where the Shin et al. approach was incapable of converging on a solution. Please see our updated explanation of this in Lines 796-805.

Ln739: In the model, the authors assume that specific productivity is proportional to the specific growth rate. But isn’t this violated in the main findings?

We thank the reviewer for pointing out this error in the way the model assumptions were presented. Indeed, the way the equations were written in the previous version of the manuscript, only those species j with nonzero a_{ij} values could modify the metabolite production per unit biomass of species i . This would exclude interactions such as those between DP and AC where there is minimal impact on growth but substantial impact on butyrate production. We had neglected to explain that the proportionality of production to growth (ρ_i) can also be a function of the abundance of other species. With similar simplifying assumptions, this results in an identical regression model, but with a more logical explanation of how the regression model parameters are related to the underlying microbial behavior that is consistent with our findings. Please see our updated explanation in Lines 839-861.

TYPOGRAPHICAL

Ln703: Figure 2a -> Figure 1c ?

Figure 2: 3-5 BPB -> 3-5 BP ?

Thank you for catching these typographical errors. They have been corrected.

Reviewer #2 (Remarks to the Author):

In this manuscript, Clark et al. build and iteratively train a bottom-up model that is able to predict butyrate production in simplified communities of Human gut bacteria. Using this model, they identify factors that affect butyrate production, both by modulating the abundance of butyrate producers, and their per capita production.

This is a very interesting and well-executed study that provides an impressive example of successfully predicting microbial community function, which is an important and timely subject with important implications for managing natural communities and designing artificial ones. It is especially notable that the authors are able to accurately predict butyrate production in communities composed of as many as 25 species - previous research in the field typically focused on communities composed of ~10 species, and predictability typically deteriorates with increasing richness. I only have a few comments that are mainly about clarity and presentation, as detailed below.

Specific comments

I found it difficult to understand exactly what are the proposed benefits of the Design-Test-Learn (DTL) paradigm in this context, and how it should be utilized. The fact that the models based on more measurement, including previously unmeasured interaction, outperform the initial simple models is not surprising. For example, The DTL paradigm can be beneficial if it can be used to guide the selection of the most informative measurements following an initial set of measurements. I'm sure that was the case in this work. If so, how were future measurements chosen? And were they more informative than an equivalent number of "naïve" measurements? It would be useful for the authors to provide general guidance to readers regarding the best way to implement the DTL paradigm when building a predictive model of community function. To clarify, this work is very valuable even if in this case the DTL approach is not more predictive than a model based on "one-shoot" measurements. But if the authors want to highlight this approach as an important aspect of the work they need to better demonstrate its utility, and explain how it was employed.

We appreciate the reviewer's interest in the efficacy of the DTL approach and we have carried out new computational efforts to clarify the benefits of this approach in our work:

- 1. We modified language to clarify our result that the communities designed in the second cycle of our DTL method (based on 1-5 species observations) had a higher variance in butyrate production than a similar set of random (i.e. "naïve") communities (Lines 243-247). This important result demonstrates that our model enabled us to design a set of communities with a certain objective (i.e. design both high and low communities, resulting in higher variance than a random set of communities).*
- 2. We performed new modeling analyses to quantify the amount of information contained in the measurements of the model designed community dataset versus the random community dataset. To do so, we trained two new models on datasets containing the 1-5 species measurements and either a randomly selected 80% of the designed communities (model M5) or a randomly selected 80% of the random communities (model M6). The M5 model informed by the designed communities outperformed the M6 model informed by the random communities in predicting the remaining 20% of communities in both datasets (same validation dataset), demonstrating that our DTL*

approach can produce more efficient experimental designs than “naïve” or random community choices. We added this result to the main text (Lines 360-372) with new supplementary figures.

Relatedly, there is some ambiguity regarding the utility of measurement of multispecies communities vs species pairs. The results section suggests that the utility of high-richness communities measurements is that they enable constructing models using fewer measurements (ll. 296-9). I interpreted this statement as saying that species indeed predominantly interact in a pairwise manner, with few higher-order interactions (HOI), but these pairwise interaction strengths can be estimated more efficiently from measurement of high-richness communities rather than all pairwise measurements. Related to the above point regarding the DTL paradigm, it's not clear whether that is true in this study (whether the measured higher-richness communities are chosen rationally or not), since the total number of measurements used to train the last model (M3) actually exceeds that number required to exhaustively measure all species pairs.

We thank the reviewer for requesting additional clarification of these points. Our models only include pairwise interaction parameters, so one possible explanation is that higher richness communities provide improved statistical power of these pairwise interactions given the same number of community measurements due to increased number of instances where a given pair of species is present in the same community.

This comment inspired us to more directly investigate the utility of pairwise measurements versus higher richness community measurements on the model's predictive capabilities by carrying out new computational efforts. To do so, we trained a new model (Model M4) informed only by observations of single species and the random high richness communities containing all 5 butyrate producers (82 11-16 species communities compared to the 110 pairwise communities included in Model M1). The new model M4 substantially outperformed Model M1 in predicting butyrate production in the other high richness communities with all 5 butyrate producers, demonstrating that random high richness communities may be a more effective initial dataset than pairwise communities for future efforts to use DTL for microbial community design. However, the high richness training set we used did not include variations in the presence of butyrate producers and thus Model M4 outperformed by Model M1 in predicting 3-5 species communities and the high richness communities lacking AC. We added a description of this analysis to the main text (Lines 339-359) and new supplementary figures showing these results.

In contrast, the discussion (second paragraph) argues that interactions actually differ in low vs high richness communities, indicating the presence of HOI. In this case, measurements of high richness communities is not a matter of inference efficiency, but rather that the effective strengths of pairwise interactions differ in these communities compared to in low richness communities. It would be valuable to clarify this point and the authors' interpretation of the actual meaning of the inferred pairwise interaction coefficients.

Upon further thought on this comment and with the additional analyses described above, we have a better picture of the importance of high richness community data for model performance. However, the interpretation of the importance of this data can take two forms. The first interpretation is that higher richness communities provide better statistical observability of pairwise interaction parameters due to increased number of instances where a given pair of species is included in the same community (e.g. for production interactions highlighted from

Line 379, only 2-10 examples containing a given pair exist in the 2-5 species experiments whereas 28-36 examples exist in the high richness communities; statistics and explanation of this point added in Lines 391-398). The second interpretation is that our system does in fact include HOI and our pairwise interaction parameters are still able to capture community behaviors when provided with high richness community data. We have added a discussion of these two points to the Discussion section (Lines 482-490, 498-504).

In panel 3f, I don't understand how is the inset consistent with the data in the main panel. In the inset, there is a clear decrease in butyrate production by AC in the presence of S within 2h. In contrast, in the panel there is no difference, or a slight increase.

We thank the reviewer for bringing up this point. In the experiment in the main figure panel, we observed decreased sulfide concentration at the end of the experiment in the AC samples relative to the other butyrate producers, which we attributed to bubbling associated with AC's fast growth. Thus, we chose to perform an additional experiment with AC with more attention to the potential loss of H₂S gas by sealing the tubes and measuring at earlier time points to observe the dynamics of butyrate production and sulfide loss. In this experiment (shown in the inset), a higher sulfide concentration was maintained, and we saw a corresponding decrease in the butyrate concentration per unit biomass. We have added an improved description of this experiment (Lines 294-302) with a new supplemental figure showing the sulfide measurements.

It was sometimes hard to find the exact number of replicates of different measurements. Adding the number of replicates in the figure captions would help.

We have added the number of replicates for each experiment to the captions.

Some of the figures are quite small and the details are hard to make out. Especially Fig. 2.

We have modified Figure 2 for clarity and visibility.

Before reading the Methods section, I was confused regarding the exact meaning and units of F. That is, that it is the butyrate concentration at a particular time point, rather than its rate of production. Perhaps edit line 122 to say "mapping community composition (i.e. abundance of each species) at a specific time point to the concentration of an output metabolite *at that time*".

Thank you for this clarifying suggestion. We have modified the text accordingly.

In the caption of Fig.1 panel (a) is labeled (b).

Thank you for catching this typographical error.

L665: were samples filtered out if they had OD600 smaller than 0.1, or larger than 0.1?

This sentence was intended to indicate that samples with few reads but measurable growth (OD600>0.1) were excluded from the analysis because the low number of reads was due to some error in sample processing rather than lack of growth. We have clarified this sentence in the text to correct this confusion (Lines 745-748).

Reviewers' Comments:

Reviewer #1:

Remarks to the Author:

All my comments have been addressed satisfactorily. The updated Figure 2 and the additional explanatory text have improved the readability of the manuscript.

Reviewer #2:

Remarks to the Author:

I thank the authors for their detailed and thoughtful response, which addressed all of my comments and further improved on an already great manuscript. I especially appreciate the new analysis involving models M4-6, which clarified and quantifies the benefits of the DTL approach, but also its limitations when there are differences in species composition between the train and test datasets.

One minor comment is that I believe that some of the text on Fig. S9b for stats of the fit for all 5 butyrate producers has been accidentally omitted.

Reviewer #1 (Remarks to the Author):

All my comments have been addressed satisfactorily. The updated Figure 2 and the additional explanatory text have improved the readability of the manuscript.

Reviewer #2 (Remarks to the Author):

I thank the authors for their detailed and thoughtful response, which addressed all of my comments and further improved on an already great manuscript. I especially appreciate the new analysis involving models M4-6, which clarified and quantifies the benefits of the DTL approach, but also its limitations when there are differences in species composition between the train and test datasets.

One minor comment is that I believe that some of the text on Fig. S9b for stats of the fit for all 5 butyrate producers has been accidentally omitted.

The p-value for the Pearson correlation for all 5 butyrate producers was 0.9 and thus not statistically significant ($p > 0.05$). Thus, we chose to include the p-value instead of the correlation coefficient and slope/intercept of the linear regression. We have added a note to the figure caption to clarify this.